# Control of complement-induced inflammatory responses to SARS-CoV-2 infection by anti-SARS-CoV-2 antibodies

Marta Bermejo-Jambrina [1,2,3,8]✉, Lieve EH van der Donk[1,2], John L van Hamme[1,2], Doris Wilflingseder[3], Godelieve de Bree[2,4], Maria Prins[4,5], Menno de Jong[6], Pythia Nieuwkerk[2,5,7], Marit J van Gils[6], Neeltje A Kootstra[1,2] & Teunis BH Geijtenbeek[1,2,8]✉

## Abstract

**Dysregulated immune responses contribute to the excessive and uncontrolled inflammation observed in severe COVID-19. However, how immunity to SARS-CoV-2 is induced and regulated remains unclear. Here, we uncover the role of the complement system in the induction of innate and adaptive immunity to SARS-CoV-2. Complement rapidly opsonizes SARS-CoV-2 particles via the lectin pathway. Complement-opsonized SARS-CoV-2 efficiently induces type-I interferon and pro-inflammatory cytokine responses via activation of dendritic cells, which are inhibited by antibodies against the complement receptors (CR) 3 and 4. Serum from COVID-19 patients, or monoclonal antibodies against SARS-CoV-2, attenuate innate and adaptive immunity induced by complement-opsonized SARS-CoV-2. Blocking of CD32, the FcγRII antibody receptor of dendritic cells, restores complement-induced immunity. These results suggest that opsonization of SARS-CoV-2 by complement is involved in the induction of innate and adaptive immunity to SARS-CoV-2 in the acute phase of infection. Subsequent antibody responses limit inflammation and restore immune homeostasis. These findings suggest that dysregulation of the complement system and FcγRII signaling may contribute to severe COVID-19.**

**Keywords** SARS-CoV-2; Complement; Dendritic Cells; Type-I IFN Responses; COVID-19
**Subject Categories** Immunology; Microbiology, Virology & Host Pathogen Interaction

## Introduction

Since severe acute respiratory syndrome coronavirus 2 (SARS-CoV-2) was first identified in Wuhan, China, in December of 2019 (Gorbalenya et al, 2020; Zhou et al, 2020a) the virus has spread all over the world, causing a respiratory disease, termed coronavirus disease 2019 (COVID-19) (World Health Organization, 2023). To date, the COVID-19 pathogenesis is still unclear. Asymptomatic patients and patients with mild COVID-19 gain control of infection within a couple of days most likely via innate immune responses as effective adaptive immune responses are expected to be elicited after 2 weeks in naive individuals (Thevarajan et al, 2020; Weitz et al, 2020). Failure of antiviral innate responses to control infection might lead to uncontrolled viral replication in the airways eliciting an inflammatory cascade observed in severe COVID-19 cases (Boechat et al, 2021; Zhang et al, 2022). Severe to fatal outcomes in COVID-19 patients have been attributed to the dysfunction of innate and adaptive immune response by SARS-CoV-2 (Arish et al, 2023). These aberrant or uncontrolled innate and/or adaptive immune responses lead to delayed viral clearance, inflammation and tissue and organ damage (Arish et al, 2023; Lopes-Pacheco et al, 2021; Yang et al, 2020). It remains unclear how the interplay between innate and adaptive immunity controls infection and how homeostasis is achieved after infection to prevent aberrant systemic inflammatory responses observed in severe COVID-19 disease.

The complement system constitutes an important innate immune response and acts as a first line of defense against viruses and might have a critical role in COVID-19 pathogenesis (Afzali et al, 2022; Lim et al, 2023; Tierney et al, 2022; Yu et al, 2022). Complement activation limits SARS-CoV-2 infection but uncontrolled activity leads to aberrant inflammatory responses observed during severe COVID-19 (Afzali et al, 2022; Ma et al, 2021; Mastellos et al, 2020). SARS-CoV-2 activates complement by direct interaction of spike proteins with the lectin pathway via

[1]Department of Experimental Immunology, Amsterdam UMC location AMC, Amsterdam, The Netherlands. [2]Amsterdam Institute for Infection and Immunity, Infectious Diseases, Amsterdam, The Netherlands. [3]Institute of Hygiene and Medical Microbiology, Medical University of Innsbruck, Innsbruck, Austria. [4]Department of Internal Medicine, Amsterdam UMC location AMC, University of Amsterdam, Amsterdam, The Netherlands. [5]Department of Infectious Diseases, Public Health Service of Amsterdam, GGD, Amsterdam, The Netherlands. [6]Department of Medical Microbiology and Infection Prevention, Amsterdam UMC location AMC University of Amsterdam, Amsterdam, The Netherlands. [7]Department of Medical Psychology (J3-2019-1), Amsterdam UMC location AMC University of Amsterdam, Amsterdam, The Netherlands. [8]These authors contributed equally: Marta Bermejo-Jambrina, Teunis BH Geijtenbeek. ✉E-mail: m.bermejojambrina@amsterdamumc.nl; t.b.geijtenbeek@amsterdamumc.nl

mannose-binding lectin (MBL) (Ali et al, 2021; Gao et al, 2022; Satyam et al, 2021; Stravalaci et al, 2022) or by binding to cell surface heparan sulfates and thereby activating the alternative pathway (Lo et al, 2022; Yu et al, 2020). SARS-CoV-2-specific antibodies binding to spike protein can also activate complement by the classical pathway through C1q (Jarlhelt et al, 2021; Lamerton et al, 2022). Severe COVID-19 patients have high circulating C5a in their blood as well as high levels of processed C3 (Mastellos et al, 2020; Skendros et al, 2022), suggesting that uncontrolled complement activation might be involved in the severity of COVID-19 (Afzali et al, 2022; Posch et al, 2021). These studies suggest that although the complement system is vital in limiting SARS-CoV-2 infection, dysregulation or lack of control of complement activation leads to severe pathogenesis (Holter et al, 2020; Java et al, 2020; Posch et al, 2021; Yu et al, 2022). Mechanisms underlying complement-induced immunity and subsequent return to homeostasis after complement activation remain unclear.

Activation of mucosal dendritic cells (DCs) is a crucial step in the induction of effective innate and adaptive immune responses against invading viruses (Soloff and Barratt-Boyes, 2010). Notably, SARS-CoV-2 infection does not lead to strong DC activation (Pérez-Gómez et al, 2021; van der Donk et al, 2022a; Zhou et al, 2020b). Exposure of DCs to SARS-CoV-2 does neither lead to infection nor induction of type-I IFN and cytokine responses. Although infection of bystander cells with SARS-CoV-2 can lead to DC activation (Jamal et al, 2021; Sanche et al, 2022; van der Donk et al, 2022a), it is unclear whether complement deposition on SARS-CoV-2 can induce DC activation.

Here, we investigated the role of complement in the induction of immunity and how the inflammatory responses are controlled to prevent aberrant inflammation. Complement-opsonized SARS-CoV-2-induced DC maturation and efficient type-I IFN responses via complement receptors CR3/CD11b and CR4/CD11c. Moreover, complement-opsonized SARS-CoV-2 induced pro-inflammatory cytokines, including inflammasome-mediated IL-1β. Notably, serum from COVID-19 patients as well as anti-SARS-CoV-2 antibodies abrogated complement-induced DC activation and subsequent type-I IFN and cytokine responses via CD32/FcγRII activation. These data strongly suggest that complement is important in induction of innate and adaptive immunity but that antibody responses either elicited after infection or vaccination, suppress complement-induced immunity and restore homeostasis. Our data imply that dysregulation in complement and FcγRII signaling might underlie mechanisms causing severe COVID-19.

# Results

## Complement-opsonized SARS-CoV-2 (hCoV-19/Italy-WT) activates dendritic cells via CR3/CD11b and CR4/CD11c

We investigated whether complement affects the interaction of SARS-CoV-2 with DCs. Incubation of pseudotyped SARS-CoV-2 and SARS-CoV-2 isolate (hCoV-19/Italy-WT) with pre-COVID-19 pandemic normal human serum (NHS), negative for SARS-COV-2-specific antibodies, led to complement opsonization of SARS-CoV-

2 as shown by the detection of C3c and C3d on virus particles (Fig. 1A,B), whereas no IgG deposition was observed (Fig. 1A). C3 depletion (C3$^{-/-}$) or heat-inactivation (HI) of sera resulted in low to no C3 deposition on SARS-CoV-2 (Fig. 1B). Moreover, C3 deposition was low on pseudovirus particles lacking SARS-CoV-2 Spike glycoprotein compared to pseudovirus particles expressing SARS-CoV-2 Spike glycoprotein (Fig. EV1A). These data strongly suggest that SARS-CoV-2 Spike glycoprotein triggers complement opsonization of SARS-CoV-2.

Notably, binding of complement-opsonized SARS-CoV-2 isolate (hCoV-19/Italy-WT) to DCs was higher than that of non-opsonized SARS-CoV-2 (Fig. 1C). Blocking antibodies against complement receptors CR3/CD11b and CR4/CD11c abrogated complement-opsonized SARS-CoV-2 binding to DCs (Fig. 1C). The lower binding of complement-opsonized SARS-CoV-2 in presence of the blocking CD11b/c antibodies than observed for SARS-CoV-2 alone, suggests that complement opsonization prevents SARS-CoV-2 binding to heparan sulfate proteoglycans (Bermejo-Jambrina et al, 2021). C3b was detected on DCs incubated with complement-opsonized SARS-CoV-2 in contrast to DCs incubated with either SARS-CoV-2 alone, exposed to C3-depleted or heat-inactivated (HI) sera (Fig. EV1B). These data strongly suggest that SARS-CoV-2 is opsonized by C3b, C3c, and C3d, and complement-opsonized SARS-CoV-2 efficiently interacts with complement receptor expressing DCs.

We next investigated the induction of DC maturation by complement-opsonized SARS-CoV-2 isolate (hCoV-19/Italy-WT) by analyzing expression of co-stimulatory molecules CD80 and CD86, DC-SIGN (Fig. 1D–I) and complement receptors CR3/CD11b and CR4/CD11c (Fig. EV1C,D). In contrast to SARS-CoV-2, complement-opsonized SARS-CoV-2 induced significant expression of CD80 (Fig. 1D,G) and CD86 (Fig. 1E,H) to similar levels as observed for LPS. Neither SARS-CoV-2 treated with C3-depleted nor HI-sera-induced CD80 or CD86 expression on DCs (Fig. EV2A–C). Upregulation of CD80 and CD86 by complement-opsonized SARS-CoV-2 was abrogated by antibodies against CR3/CD11b and CR4/CD11c alone or in combination (Fig. 1D,E), whereas isotype antibodies did not affect DC activation (Fig. EV2D–F). DC-SIGN expression was reduced by complement-opsonized SARS-CoV-2 (Fig. 1F,I), similar as observed for LPS, and expression was restored in the presence of blocking antibodies against CR3/CD11b and/or CR4/CD11c (Fig. 1F). Moreover, NHS alone did not activate DCs (Fig. EV2G). Opsonization of SARS-CoV-2 and subsequent DC activation was concentration-dependent (Fig. EV3A–C). Neither non-opsonized nor complement-opsonized SARS-CoV-2 productively infected DCs as no virus production was observed over time (Fig. 2A,B) (Bermejo-Jambrina et al, 2021; van der Donk et al, 2022b), suggesting that virus replication is not involved in DC activation. These results strongly suggest that complement opsonization enhances SARS-CoV-2 capture by DCs and induces DC maturation via CR3 and CR4.

## Complement-opsonized SARS-CoV-2 (hCoV-19/Italy-WT) induces type-I IFN and cytokine responses

Next, we investigated whether complement-opsonized SARS-CoV-2 isolate (hCoV-19/Italy-WT) induces antiviral type-I interferon (IFN) as well as cytokine responses. Notably, in contrast to non-opsonized SARS-CoV-2, complement-opsonized SARS-CoV-2

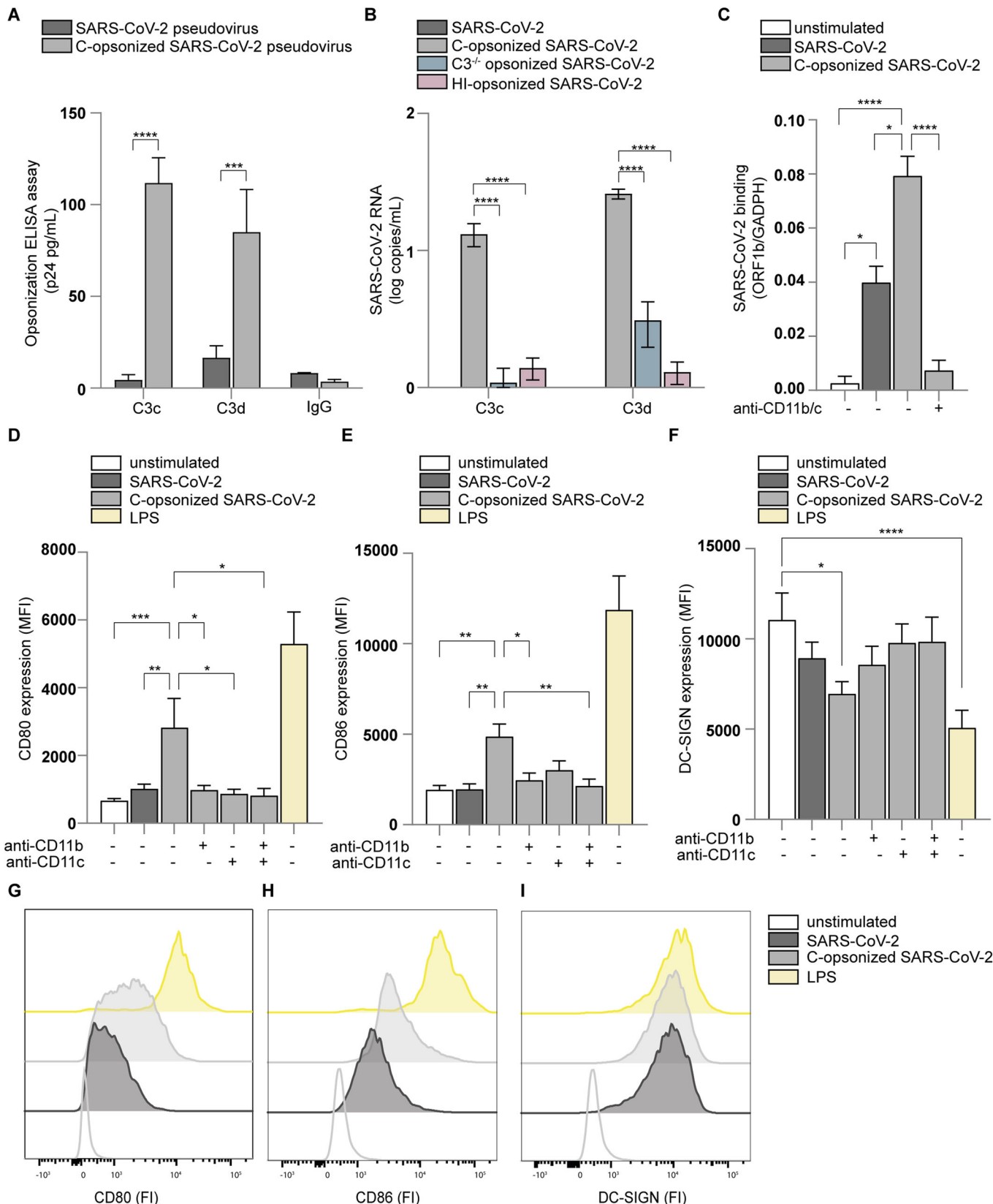

**Figure 1. Complement-opsonized SARS-CoV-2 activates DCs via CD11b and CD11c.**

(A, B) SARS-CoV-2 pseudovirus (A) and SARS-CoV-2 isolate (hCoV-19/Italy-WT, 1000 TCID/mL) (B) opsonization by C3c and C3d were determined by (A) ELISA (p24 pg/mL) or (B) qPCR, respectively ($n = 3$ donors). (C) Human monocyte-derived DCs were exposed to SARS-CoV-2 isolate (hCoV-19/Italy-WT, 1000 TCID/mL) and complement-opsonized SARS-CoV-2 (hCoV-19/Italy-WT, 1000 TCID/mL) for 4 h at 4 °C in presence or absence of antibodies against CD11b and CD11c. Virus binding was determined by quantitative real-time PCR ($n = 6$ donors). (D–F) DCs were exposed to SARS-CoV-2 or complement-opsonized SARS-CoV-2 for 24 h in the presence or absence of antibodies against CD11b and CD11c and expression of CD80, CD86 and DC-SIGN was determined by flow cytometry ($n = 12$ donors). LPS stimulation was used as a positive control. (G–I) Representative histograms of CD80 (G), CD86 (H), and DC-SIGN (I) expression. Data show the mean values and error bars are the SEM. Statistical analysis was performed using (A, B) ordinary one-way ANOVA with Tukey multiple-comparison test. ***$P \leq 0.001$, ****$P \leq 0.0001$ ($n = 3$ donors). (C) Ordinary one-way ANOVA with Tukey multiple-comparison test. **$P \leq 0.01$, ***$P \leq 0.001$, ****$P \leq 0.0001$ ($n = 6$ donors). (D–F) Two-way ANOVA with Tukey multiple-comparison test. *$P \leq 0.05$, **$P \leq 0.01$, ***$P \leq 0.001$, ****$P \leq 0.0001$ ($n = 12$ donors). Source data are available online for this figure.

isolate (hCoV-19/Italy-WT) induced significantly higher mRNA levels of IFNβ as well as IFN-stimulated genes (ISGs) APOBEC3G, IRF7, and CXCL10 (Fig. 2C–F). Blocking antibodies against CR3/CD11b and CR4/CD11c, in contrast to isotype antibodies, abrogated induction of IFNβ and ISGs to similar levels as observed with SARS-CoV-2 alone (Figs. 2C–F and EV3E–G). Moreover, no type-I IFN response was induced by SARS-CoV-2 treated with C3-depleted or HI-sera (Fig. EV3E–G). These data strongly suggest that in contrast to non-opsonized virus, complement-opsonized SARS-CoV-2 induces antiviral type-I IFN responses via CR3 and CR4.

Moreover, complement-opsonized SARS-CoV-2 induced transcription of cytokines IL-6, IL-10, and IL-12p35, and expression was abrogated by blocking CR3/CD11b and CR4/CD11c (Fig. 2G–I). Secretion of biologically active IL-1β is tightly regulated and depends on induction of pro-IL-1β and caspase-1-dependent processing into IL-1β, which is subsequently secreted by DCs (Franchi Muñoz-Planillo and Núñez, 2012; Martinon Mayor and Tschopp, 2009; Schroder and Tschopp, 2010). Notably, complement-opsonized SARS-CoV-2 induced secretion of IL-1β protein in contrast to non-opsonized SARS-CoV-2, and IL-1β production was inhibited by antibodies against CR3/CD11b and CR4/CD11c (Fig. 3A) in contrast to isotype antibodies (Fig. EV3G). Complement-opsonized SARS-CoV-2 significantly induced caspase-1 activity in DCs, similar as observed for LPS- and ATP-stimulated DCs (Figs. 3B and EV3C). Caspase-1 activation by complement-opsonized SARS-CoV-2 was blocked by antibodies against CR3/CD11b and CR4/CD11c. Non-opsonized SARS-CoV-2 did not activate caspase-1 (Fig. 3B).

Next, we investigated whether the lectin pathway was involved in complement activation by SARS-CoV-2. Pre-treatment of pre-COVID-19 pandemic NHS with mannan, which act as an inhibitor of the lectin pathway via MBL (Héja et al, 2012), significantly decreased DC-induced type-I IFN and IL-6 responses (Figs. 3C–E and EV4A,B), indicating that SARS-CoV-2 activates complement by the lectin pathway through the carbohydrate-recognition domain. Together these data strongly suggest that complement opsonization of SARS-CoV-2 by MBL induces a potent pro-inflammatory as well as an antiviral type-I IFN response in DCs via CR3 and CR4.

## Convalescent serum from COVID-19 patients blocks immune responses induced by complement-opsonized SARS-CoV-2 isolate via CD32/FcγRII

Antibodies are important in the induction of complement activation and subsequent deposition (Goldberg and Ackerman, 2020; Sörman et al, 2014). Here, we investigated whether antibodies against SARS-CoV-2 affect complement-induced immunity by DCs. Serum from 20 (mild/moderate disease) recovered COVID-19 patients neutralized SARS-CoV-2 infection, indicating that serum contains neutralizing antibodies against SARS-CoV-2 (Brouwer et al, 2020; Caniels et al, 2021b; van Gils et al, 2022a). Next, we investigated the effect of COVID-19 serum on complement-induced immunity. Complement-opsonized SARS-CoV-2 (hCoV-19/Italy-WT) induced IFNβ, APOBEC3G, IRF7 and CXCL10 as well as cytokines IL-6, IL-10 and IL-12p35 (Fig. 4A–G). Notably, pre-incubation of complement-opsonized SARS-CoV-2 with COVID-19 convalescent serum abrogated type-I IFN and cytokine responses and responses were restored by blocking FcγRII (CD32) (Fig. 4A–G). These data strongly suggest that antibodies against SARS-CoV-2 suppress immune responses induced by complement-opsonized SARS-CoV-2 via CD32/FcγRII .

## Monoclonal antibodies against SARS-CoV-2 block complement-induced immunity to SARS-CoV-2

We investigated whether monoclonal antibodies against SARS-CoV-2 suppress the inflammation induced by complement-opsonized SARS-CoV-2 and how this is affected by the neutralizing capacity. We compared the effect of non-neutralizing and neutralizing antibodies against SARS-CoV-2 isolated from COVID-19 patients, COVA1-27 and COVA1-18, respectively (Brouwer et al, 2020). Complement-opsonized SARS-CoV-2 induced IFNβ transcription and, notably, pre-incubation of complement-opsonized SARS-CoV-2 with either neutralizing (COVA1-18) or non-neutralizing antibody (COVA1-27) against SARS-CoV-2 abrogated IFNβ transcription (Fig. 5A). The COVA1-27-mediated suppression was abrogated by CD32/FcγRII inhibition thereby restoring IFNβ transcription to levels observed with complement-opsonized SARS-CoV-2 (Fig. 5A). CD32/FcγRII inhibition had less effect on COVA1-18-mediated suppression (Fig. 5A). Similarly, both COVA1-18 and 1-27 suppressed induction of ISGs APOBEC3G, IRF7 and CXCL10 as well as cytokines IL-6, IL-10, and IL-12p35 (Fig. 5B–G). CD32/FcγRII inhibition restored induction of CXCL10 and partially for APOBEC3G, IRF7, and IL-12p35. In contrast to IL-6, IL-10 induction was restored by CD32/FcγRII inhibition. Moreover, induction of CD86 was also suppressed by COVA1-18 and 1-27, which was restored by CD32/FcγRII inhibition (Fig. EV4C). These data strongly suggest that antibodies against SARS-CoV-2 present in serum inhibit complement-induced immune responses partially or completely via CD32/FcγRII, and this is independent of neutralization capacity.

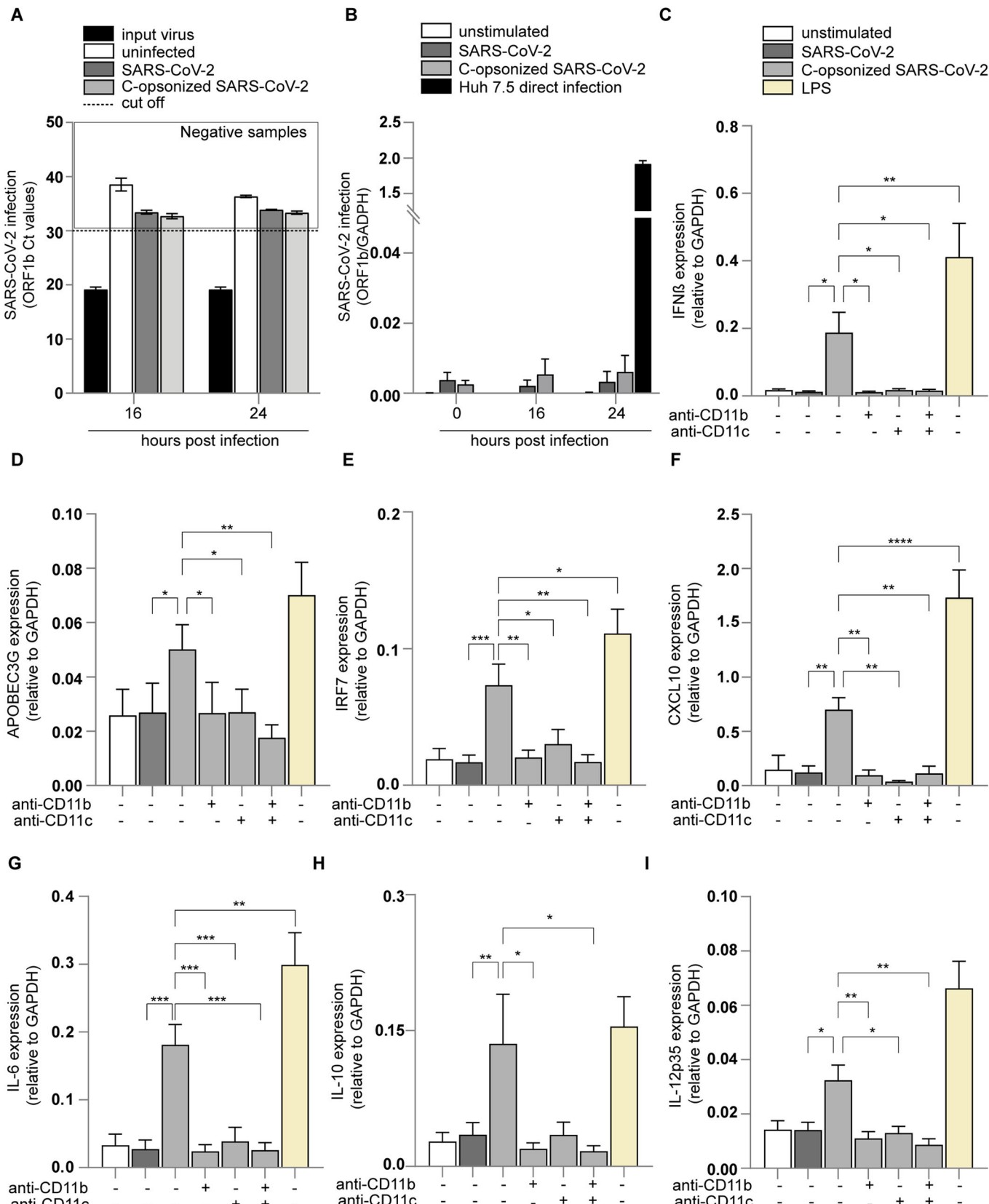

**Figure 2.  Complement-opsonized SARS-CoV-2 induces type-I IFN and cytokine responses via CRs.**

(A, B) ACE2-negative human monocyte-derived DCs were exposed to SARS-CoV-2 isolate (hCoV-19/Italy-WT, 1000 TCID/mL) and complement-opsonized SARS-CoV-2 (hCoV-19/Italy-WT, 1000 TCID/mL) for 16 h and 24 h. Viral production detectable in the supernatant (A) and cell infection ($n = 2$ donors) (B) were determined by qPCR. (C–I) DC were exposed to SARS-CoV-2 isolate (hCoV-19/Italy-WT, 1000 TCID/mL), complement-opsonized SARS-CoV-2 (hCoV-19/Italy-WT, 1000 TCID/mL) and LPS (100 ng/mL) in presence or absence of antibodies against CD11b and CD11c for 2 h and 6 h. mRNA levels of IFN-β (C), APOBEC3G (D), IRF7 (E), CXCL10 (F), IL-6 (G), IL-10 (H), and IL-12p35 (I) were determined with qPCR after 2 h (C) and after 6 h (D–I) ($n = 12$ donors). Data show the mean values and error bars are the SEM. Statistical analysis was performed using (C–I) two-way ANOVA with Dunnett's multiple-comparison test. *$P \le 0.05$, **$P \le 0.01$, ***$P \le 0.001$, ****$P \le 0.0001$ ($n = 12$ donors). Source data are available online for this figure.

## Serum samples from mild and severe COVID-19 patients block complement-induced immunity to SARS-CoV-2

Circulating immune complexes have been correlated with complement activation in severe/critical COVID-19 patients (Caniels et al, 2021a; van Gils et al, 2022a; Verveen et al, 2022; Wynberg et al, 2022). To analyze the impact of antibody status and complement function in parallel, we screened serum from 5 individuals with mild outcome versus 5 severe COVID-19. Serum from both groups were incubated for 1 h at 37 °C with SARS-CoV-2 isolate (hCoV-19/Italy-WT) and the presence of C3c/C3d and IgGs was determined by ELISA (Fig. 6A). Opsonization of SARS-CoV-2 with serum from either mild or severe COVID-19 patients led to the deposition of C3c and C3d fragments as well as IgG on SARS-CoV-2. Serum from mild COVID-19 patients caused inferior opsonization by C3c/d compared to IgG, whereas serum from severe COVID-19 patients induced more C3c/d opsonization compared to IgG. We therefore investigated whether antibodies present in sera did not alter the opsonization. We opsonized SARS-CoV-2 in the presence of pre-COVID-19 pandemic NHS together with COVA1-18 mAb. The complement deposition (C3c and C3d) did not change due to the presence of the antibody (Fig. EV5A). We further analyzed the levels of complement activation products, C3a, and C5a, in serum samples from mild ($n = 7$) and severe ($n = 7$) COVID-19 and healthy donors (HD). Notably, C3a and C5a were significantly elevated in severe COVID-19 patients in comparison to mild and healthy donors (Fig. EV5B,C). These results suggest that in severe COVID-19 patients, complement is fully activated. DCs from healthy donors were stimulated with opsonized SARS-CoV-2 with serum from either mild or severe COVID-19 patients and induction of IFNβ, ISGs, such as IRF7, and IL-6 was measured (Fig. 6B–D). Serum from both COVID-19 mild and severe patients suppressed IRF7 and IL-6 induction, which was restored by CD32/FcγRII inhibition (Fig. 6B–D). There was a trend observed for suppression of IFNβ induction by sera from severe patients SARS-CoV-2, which was restored although not significantly by CD32/FcγRII inhibition. Interestingly, CD32/FcγRII inhibition enhanced both type-I IFN and IL-6 responses compared to complement induction alone, which might be due to higher complement concentrations in serum from severe COVID-19 patients as suggested by higher opsonization of SARS-CoV-2 (Fig. 6A). SARS-CoV-2 opsonized with sera from mild patients induced APOBEC3G and CXCL10, which was inhibited by CD32/FcγRII block, whereas low induction of IL-10 and IL-12p35 was observed (Fig. EV5D–G). Notably, induction of APOBEC3G by sera from severe patients SARS-CoV-2 was almost restored by CD32 inhibition, while there is a minimal or absent response in sera from severe patients for CXCL10, IL-10 and IL-12p35

(Fig. EV5D–G). Although DCs expressed CD16 (FcγRIII) (Fig. EV4E), CD16 inhibition did not affect the suppression of IFNβ induction (Fig. EV4D). These data suggest that the generation of antibodies during COVID-19 disease are important to resolve inflammation.

## Discussion

Complement is crucial for the induction of inflammatory responses to pathogens leading to an effective adaptive immune response. A hallmark of severe COVID-19 disease is excessive inflammation associated with enhanced morbidity and mortality. Accumulating evidence suggests that overactivation of the complement system contributes to the pathophysiology of severe COVID-19 disease. Previously, we have shown that authentic SARS-CoV-2 isolate (hCoV-19/Italy-WT) (1000 TCID/mL or MOI 0.028) do not activate DCs, which suggests immune escape (van der Donk et al, 2022a). Here we show that SARS-CoV-2 is efficiently opsonized by complement C3b/d fragments and complement-opsonized SARS-CoV-2 efficiently induced DC activation, type-I IFN and cytokine responses via complement receptors CR3/CD11b and CR4/CD11c. Notably, we identified antibody responses against SARS-CoV-2 as a negative feedback mechanism in limiting complement-induced inflammation via CD32/FcγRII binding. Our data therefore suggest that complement is crucial in the induction of antiviral innate and adaptive immune responses to SARS-CoV-2 and subsequently elicited antibodies against SARS-CoV-2 downregulate complement-induced immunity, thereby preventing aberrant inflammation. This study highlights the important role for antibodies against SARS-CoV-2 to control immune homeostasis and suggest that dysregulation in this control might underlie aberrant inflammatory responses observed in severe COVID-19.

SARS-CoV-2 itself can activate the complement system directly through the lectin pathway (Ali et al, 2021; Gao et al, 2022; Hurler et al, 2023; Magro et al, 2020; Malaquias et al, 2021) and the alternative pathway (Yu et al, 2020). Specifically, SARS-CoV-2 Spike and nucleocapsid proteins are directly recognized by the lectin pathway components, leading to complement activation and the subsequent complement deposition (C3b) on virions (Ali et al, 2021). We observed that inoculation of SARS-CoV-2 isolate (hCoV-19/Italy-WT) with human pre-COVID-19 pandemic serum led to efficient opsonization of SARS-CoV-2 by C3c and C3d fragments. Opsonization was dependent on C3 as C3 depletion-abrogated opsonization. Moreover, the SARS-CoV-2 Spike glycoprotein triggered the complement activation as Spike-negative virus particles did not induce opsonization. Opsonization was inhibited by carbohydrate mannan, strongly supporting a role for MBL and

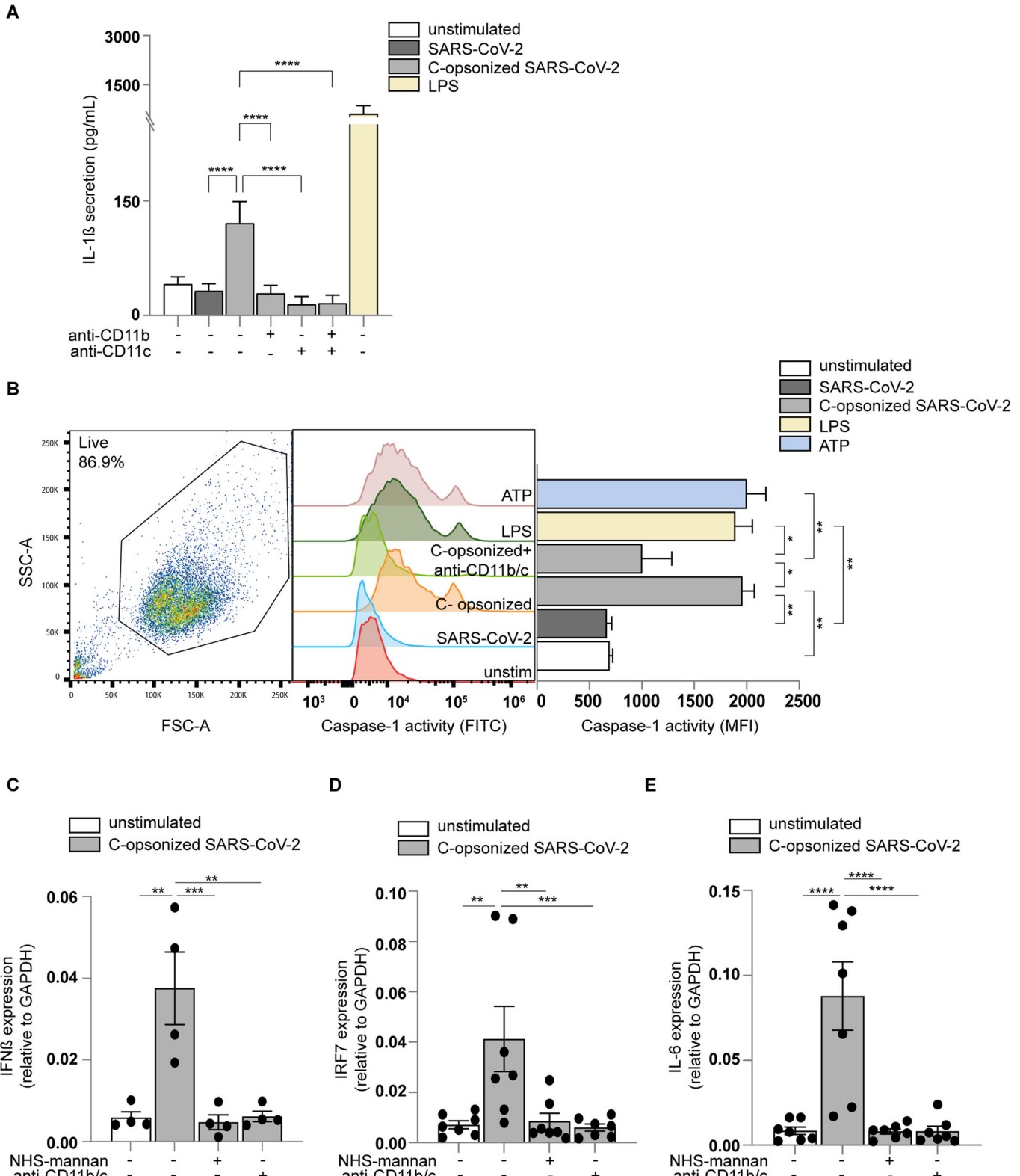

Figure 3. **Complement-opsonized SARS-CoV-2 induces caspase-1 medited IL-1β secretion.**

(A) Human monocyte-derived DC were exposed to SARS-CoV-2 isolate (hCoV-19/Italy-WT, 1000 TCID/mL), complement-opsonized SARS-CoV-2 (hCoV-19/Italy-WT, 1000 TCID/mL) and LPS (100 ng/mL) in presence or absence of antibodies against CD11b and CD11c and IL-1β secretion (pg/mL) in the supernatant was measured after 24 h by ELISA (n = 4 donors). (B) DCs were left unstimulated or treated with anti-CD11b/c prior to exposure to non- and complement-opsonized SARS-CoV-2, LPS and ATP. DCs with active caspase-1 were detected after 14 h by flow cytometry using the FAM-FLICA assay (n = 3 donors). (C–E) NHS was incubated with mannan, prior SARS-CoV-2 opsonization. DCs were exposed to non-, complement-opsonized SARS-CoV-2 and NHS-mannan opsonized SARS-CoV-2 in the presence or absence of anti-CD11b/c, and mRNA levels of IFNβ (n = 4 donors) (C), IRF7 (n = 7 donors) (D), and IL-6 (n = 7 donors) (E) were determined by qPCR. Data show the mean values and error bars are the SEM. Statistical analysis was performed using (A) two-way ANOVA with Tukey multiple-comparison test. ****P ≤ 0.0001 (n = 4 donors). (B) ordinary one-way with Tukey's multiple-comparison test. *P ≤ 0.05, **P ≤ 0.01 (n = 3 donors). (C–E) Two-way ANOVA with Tukey multiple-comparison test. **P ≤ 0.01, ***P ≤ 0.001, ****P ≤ 0.0001 (n = 4 donors) (C), (n = 7 donors) (D), and (n = 7 donors) (E). Source data are available online for this figure.

the lectin pathway in activating complement. The use of pre-COVID-19 pandemic serum excluded a potential role for antibodies against SARS-CoV-2 in complement activation as has been shown others (Gaikwad et al, 2022; Holter et al, 2020; Jarlhelt et al, 2021; Poston et al, 2021).

SARS-CoV-2 (hCoV-19/Italy-WT) binds to human DCs via heparan sulfate proteoglycans and C-type lectin receptor DC-SIGN (Bermejo-Jambrina et al, 2021; Clausen et al, 2020; Lempp et al, 2021; Thépaut et al, 2021). SARS-CoV-2 binding to DCs is important for viral transmission to epithelial cells but does not cause immune activation (Bermejo-Jambrina et al, 2021; Singh et al, 2022; van der Donk et al, 2022a). The lack of immune activation by SARS-CoV-2 is at least partially due to finding that human DCs do not become infected by SARS-CoV-2 as these immune cells do not express ACE2 (Song et al, 2023; van der Donk et al, 2022a). Complement-opsonized SARS-CoV-2 was more efficiently bound by DCs than non-opsonized SARS-CoV-2, and binding was inhibited by antibodies against complement receptors CR3/CD11b and CR4/CD11c. The enhanced binding of complement-opsonized SARS-CoV-2 to DCs is consistent with the increased uptake that has also been observed for other viruses, such as HIV-1 and HSV- 2 in previous studies (Bermejo-Jambrina et al, 2020; Crisci et al, 2016; Posch et al, 2015; Tjomsland et al, 2011).

In contrast to SARS-CoV-2 alone, complement-opsonized SARS-CoV-2 strongly induced DC maturation as determined by upregulation of the co-stimulatory molecules CD80 and CD86. Moreover, complement-opsonized SARS-CoV-2 induced expression of IFNβ and ISGs such IRF7, APOBEC3G and CXCL10 as well as cytokines IL-6, IL-10 and IL-1β. Type-I IFN responses are pivotal to antiviral immunity by induction of innate resistance to virus replication but also activating cytotoxic T-cell and T-helper cell responses to viruses (Crouse Kalinke and Oxenius, 2015; McNab et al, 2015; Park and Iwasaki, 2020). In particular, IL-1β is a very potent pro-inflammatory cytokine activating both innate and adaptive immune responses (Mantovani et al, 2019; Van Den Eeckhout Tavernier and Gerlo, 2021). Neither SARS-CoV-2 nor complement-opsonized SARS-CoV-2 productively infected DCs. Interestingly, ectopic expression of ACE2 leads to productive SARS-CoV-2 infection and replication triggers cytosolic pattern recognition receptors (van der Donk et al, 2022a) such as cytosolic RIG-I like receptors (Thorne et al, 2021), leading to DC activation. These data strongly suggest that complement-opsonized SARS-CoV-2 does not trigger endosomal TLRs similar as SARS-CoV-2 but instead the interaction with CR3 and CR4 induce DC activation and subsequent cytokine responses. Moreover, our data suggest that complement-opsonized SARS-CoV-2 binding to DCs via CR3 and CR4 leads to pro-IL-1β expression and subsequent activation

of caspase-1 inflammasome and processing of pro-IL-1β into bioactive IL-1β. Although IL-1β induction is important to induce innate and adaptive immunity, unrestrained expression of IL-1β leads to severe inflammation in different diseases (Carta et al, 2017; Hoffman and Wanderer, 2010; Kaneko et al, 2019; Potere et al, 2022). Several studies suggest that IL-1β production is an important factor in inflammatory responses during COVID-19 (Del Valle et al, 2020; Makaremi et al, 2022; Potere et al, 2022; Yudhawati Sakina and Fitriah, 2022) but mechanisms that control IL-1β production or type-I IFN responses upon SARS-CoV-2 infection remain unidentified.

Antibodies against SARS-CoV-2 have been suggested to activate complement during infection (Castanha et al, 2022; Dufloo et al, 2021; Farkash et al, 2021; Jarlhelt et al, 2021). We neither observed induction of complement opsonization of SARS-CoV-2 isolate (hCoV-19/Italy-WT) by serum from COVID-19 patients nor monoclonal human non- or neutralizing antibodies against SARS-CoV-2. In contrast, we observed that serum from COVID-19 patients as well as monoclonal human antibodies against SARS-CoV-2 attenuated complement-opsonized SARS-CoV-2-induced immune inflammatory response. The observed DC maturation as well as type-I IFN and cytokine responses induced by complement-opsonized SARS-CoV-2 was inhibited to levels observed for SARS-CoV-2 by serum from mild and severe COVID-19 patients. Similarly, both neutralizing and non-neutralizing antibodies against SARS-CoV-2 blocked these immune responses induced by complement-opsonized SARS-CoV-2.

Human DCs express both activating (FcγRIIa) and inhibitory (FcγRIIb) isoforms of FcγRII (Boruchov et al, 2005). The FcγRIIa (CD32a) isoform is involved in DC activation by IgG (Boruchov et al, 2005; Vogelpoel et al, 2015). Notably, our data strongly suggest that IgG antibodies against SARS-CoV-2 suppress complement-induced immunity by CD32/FcγRII as blocking antibodies against CD32 restored immune activation induced by complement-opsonized SARS-CoV-2. These data strongly suggest that antibodies against SARS-CoV-2 might be important in switching off complement-induced immunity and could be used to treat patients suffering from severe COVID-19 (Casadevall Joyner and Pirofski, 2020; Casadevall Pirofski and Joyner, 2021; Herman et al, 2021; Janssen et al, 2020). Moreover, our finding indicate that the inhibitory isoform FcγRIIb is involved in interacting with immune complexes of complement-opsonized SARS-CoV-2 and antibodies against SARS-CoV-2. FcγRIIb has the immunoreceptor tyrosine-based inhibitory motif (ITIM) and crosslinking blocks DC maturation (Boruchov et al, 2005; Smith and Clatworthy, 2010). FcγRIIb activation by anti-SARS-CoV-2 antibodies might prevent CD11b and CD11c signaling by dephosphorylation of kinases such as tyrosine kinase Syk and those of the Src family, which have been shown to be involved in CD11b/c signaling (Berton Mócsai and Lowell, 2005; Petty Worth and Todd, 2002).

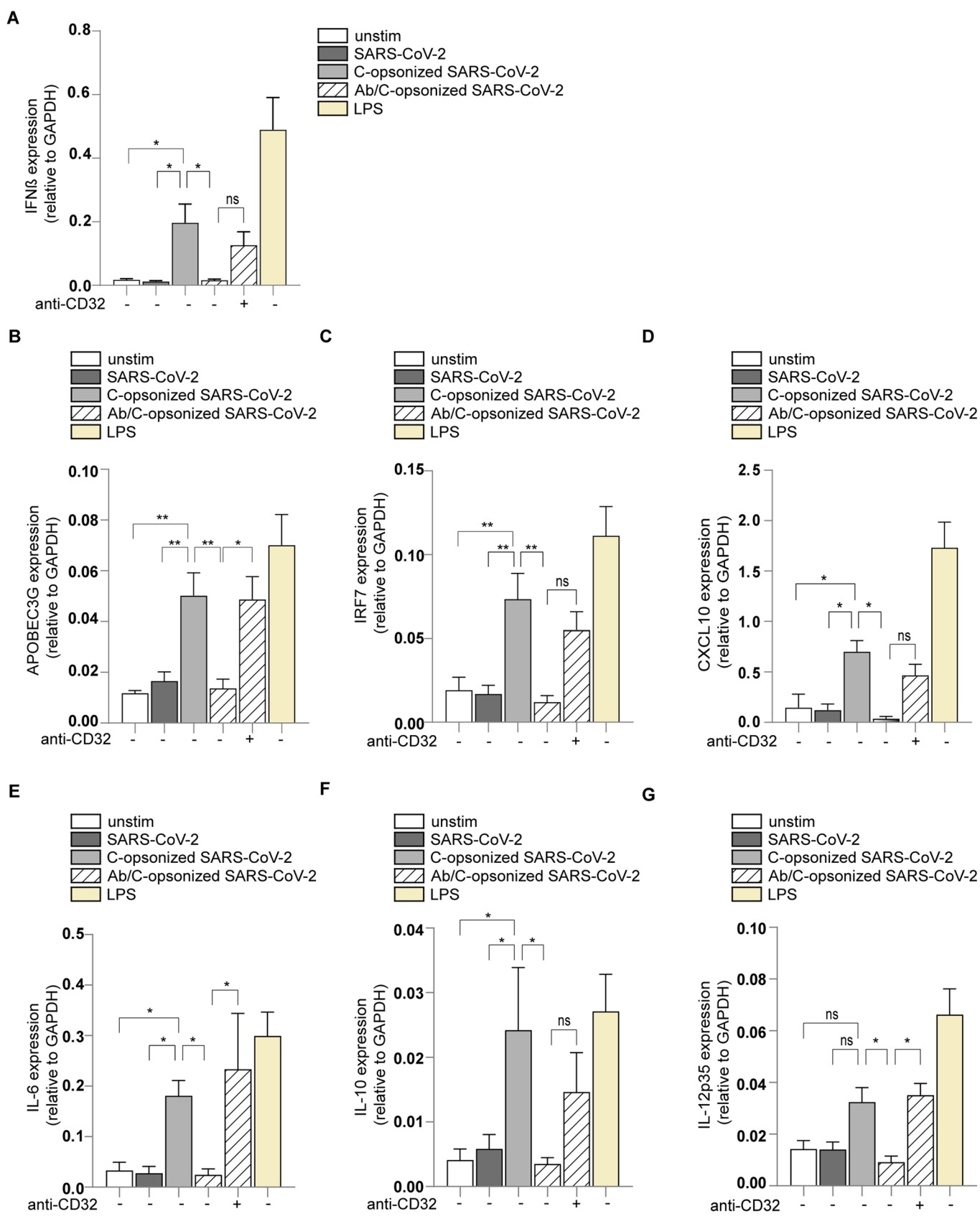

◄ Figure 4. Anti-SARS-CoV-2 antibodies present in sera suppress complement activation mediated immune activation via CD32/FcγRII.

(A–G) Human monocyte-derived DCs were exposed to SARS-CoV-2 isolate (hCoV-19/Italy-WT, 1000 TCID/mL), complement-opsonized SARS-CoV-2 (hCoV-19/Italy-WT, 1000 TCID/mL) and antibody/complement-opsonized SARS-CoV-2 (hCoV-19/Italy-WT, 1000 TCID/mL), made by a pooled sera from 20 mild COVID-19 patients supplemented with pre-pandemic NHS, as well as LPS (100 ng/mL) in presence or absence of anti-CD32 for 2 h (A) and 6 h (B–G). mRNA levels of IFNβ (A), APOBEC3G (B), IRF7 (C), CXCL10 (D), IL-6 (E), IL-10 (F), and IL-12p35 (G) were determined with qPCR after 6 h ($n = 12$ donors). Data show the mean values and error bars are the SEM. Statistical analysis was performed using (A–G) two-way ANOVA with Dunnett's multiple-comparison test. ns not significant. *$P \le 0.05$, **$P \le 0.01$ ($n = 12$ donors). Source data are available online for this figure.

Our data suggest that FcγRIIb is involved in restoring homeostasis after SARS-CoV-2 infection. Both activating and inhibitory isoforms are expressed by DCs, and the expression of both isoforms is controlled by different stimuli (Boruchov et al, 2005). A dysregulation of the balance between the FcγRII isoforms is thought to be involved in autoimmune diseases (Anania et al, 2019; Smith and Clatworthy, 2010). Therefore it will be important to investigate the expression of both FcγRII isoforms during SARS-CoV-2 infection and whether the ratio is dysregulated in individuals suffering from severe COVID-19. However, by using COVID-19 cohort samples, we demonstrated that non-heat-inactivated serum from mild and severe COVID-19 patients similarly suppressed IFNβ, ISG IRF7, and cytokine IL-6 induced by complement-opsonized SARS-CoV-2. We observed that serum from severe COVID-19 patients had enhanced C3c and C3d deposition on SARS-CoV-2 than virus opsonized by serum from mild COVID-19 patients, whereas antibody deposition was increased with serum from mild COVID-19 patients. Although serum from mild and severe patients similarly suppressed immune responses, we observed that blocking CD32/FcγRII led to an enhanced inflammatory response in the presence of serum from severe patients. These data suggest that increased complement opsonization of SARS-CoV-2 by serum from severe COVID-19 patients will lead to more severe inflammation early during infection when no antibodies are present. Several studies have reported that SARS-CoV-2-specific IgG depends on COVID-19 severity with the highest virus-specific IgG antibody titers in severe cases (Chakraborty et al, 2021; Hoepel et al, 2021; Yan et al, 2022; Yang et al, 2021) and aberrant inflammation observed in severe cases is often accompanied by high titers of specific antibodies (Hendriks et al, 2022; Jing et al, 2022). These studies suggest that antibodies against SARS-CoV-2 are present in severe cases and as such should attenuate complement-induced inflammation as we also observe with sera from patients with severe COVID-19 on DCs from healthy donors. It is possible that early during infection enhanced complement activation leads to strong immune responses leading to initiation of severe COVID-19, or that the attenuation of complement-induced inflammation by antibodies in severe cases is less efficient due to antibody glycosylation or single nucleotide polymorphisms in genes involved in CD32/FcγRII signaling. Several studies have shown that glycosylation of IgG in severe COVID-19 cases is distinct from mild cases (Chakraborty et al, 2021; Hoepel et al, 2021; Vicente et al, 2022). Moreover, SNPs in CD32/FcγRII signaling pathways have been identified and these might affect the attenuation of the complement-induced inflammation (Delidakis et al, 2022; Robinson et al, 2022).

Our data suggest that complement activation by the MBL pathway is important for the induction of innate and adaptive antiviral immunity to SARS-CoV-2 via CR3 and/or CR4 on human DCs. Complement is present in mucosal tissues, and this will lead to rapid activation of immunity upon SARS-CoV-2 infection. Our data have uncovered a striking role for antibodies against SARS-CoV-2 in attenuating the complement-induced inflammatory responses and thereby might be required in resolving inflammation. Moreover, as other studies have also shown (Nijmeijer et al, 2021; Posch et al, 2012), we demonstrate the importance of studying SARS-CoV-2 infection and other viruses under conditions that reflect the in vivo situation; viral particles covered by complement fragments and/or complement fragments and antibodies, as these factors have a profound effect on the virus interaction with the host.

Notably, our findings support a role for antibodies against SARS-CoV-2 induced by vaccinations and after natural infection, not only in limiting infection but importantly in attenuating inflammation upon SARS-CoV-2 infection. Genetic polymorphisms in CD32/FcγRII signaling pathways involved in attenuating complement-induced immunity might be responsible for unresolved inflammatory responses observed in severe COVID-19. Polymorphisms in MBL and FcγRII have been associated with susceptibility to or severity of some infectious diseases, such SARS-CoV or influenza (Fricke-Galindo and Falfán-Valencia, 2021; Mehrbod et al, 2021; Yuan et al, 2005) as well as COVID-19 (López-Martínez et al, 2022; Malaquias et al, 2021; Medetalibeyoglu et al, 2021; Stravalaci et al, 2022), but whether these affect the complement activation and negative feedback mechanism remains to be investigated. We here provide novel immunologic and mechanistic insights into SARS-CoV-2 infection, where the host can cope with the virus due to efficient cellular and humoral immune response. These findings might be exploited for future therapeutic options to improve antiviral immune responses via triggering not yet considered host mechanisms, i.e., complement receptors expressed on immune cells.

## Methods

### Ethics statement

This study was performed according to the Amsterdam University Medical Centers, location Academic Medical Center (AMC), and human material was obtained in accordance with the AMC Medical Ethics Review Committee (Institutional Review Committee) following the Medical Ethics Committee guidelines. This study, including the tissue harvesting procedures, was consent by all donors and conducted in accordance with the ethical principles set out in the declaration of Helsinki and was approved by the institutional review board of the Amsterdam University Medical Centers and the Ethics Advisory Body of the Sanquin Blood Supply Foundation (Amsterdam, Netherlands). All research was performed in accordance with appropriate guidelines and regulations.

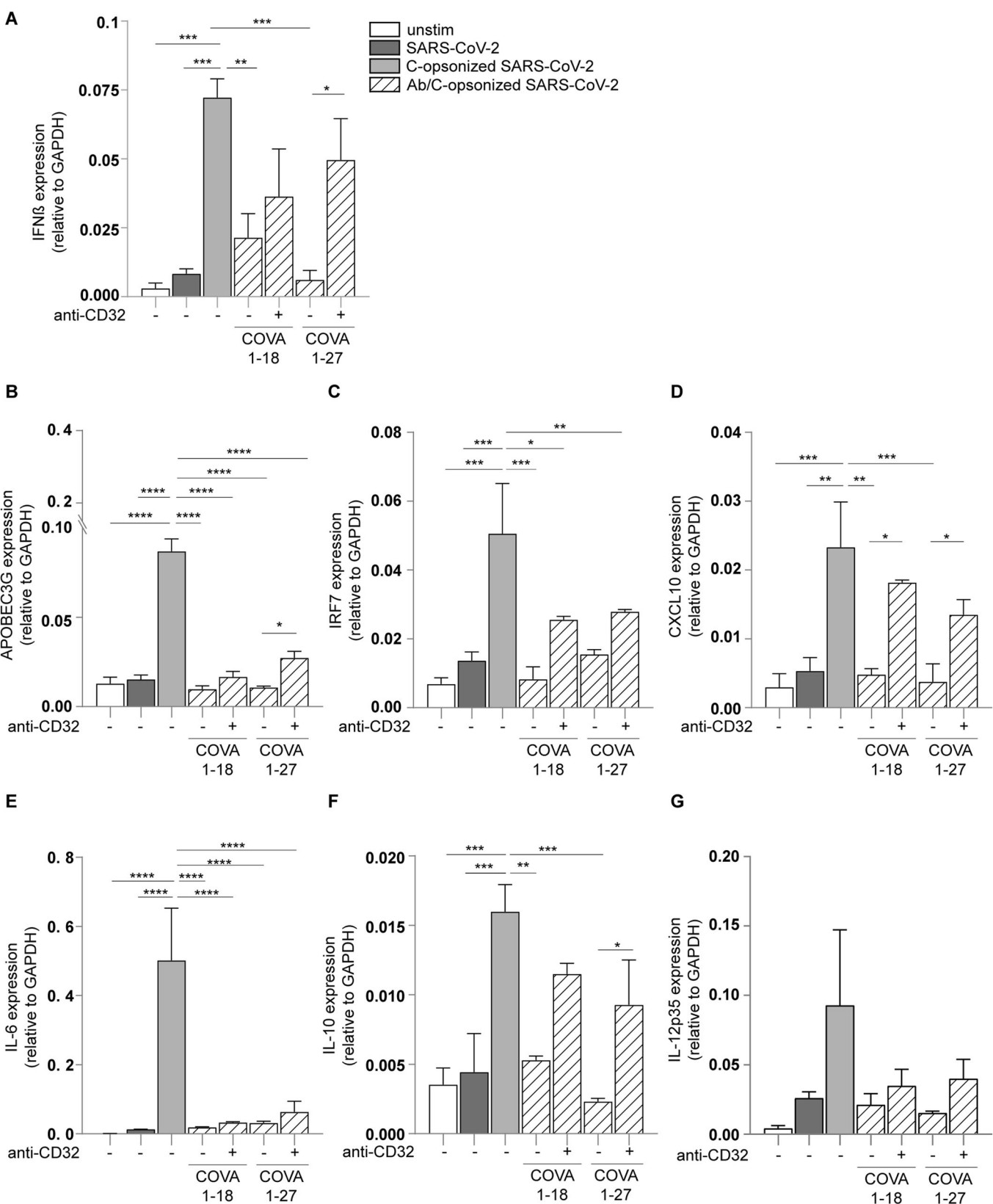

**Figure 5. Non- and neutralizing anti-SARS-CoV-2 monoclonal antibodies suppress complement activation mediated immune activation via CD32/FcγRII.**

(A–G) Pre-incubation of complement-opsonized SARS-CoV-2 with patient isolated mAb COVA1-18 (0.05 µg/mL) and COVA1-27 (0.05 µg/mL) for 30 min at 37 °C led to the antibody/complement-opsonized SARS-CoV-2 condition. Human monocyte-derived DCs were exposed to SARS-CoV-2 isolate (hCoV-19/Italy-WT, 1000 TCID/mL), to complement-opsonized SARS-CoV-2 (hCoV-19/Italy-WT, 1000 TCID/mL) and antibody/complement-opsonized SARS-CoV-2 (hCoV-19/Italy-WT, 1000 TCID/mL) in the presence or absence of anti-CD32 for 2 h and 6 h. mRNA levels of IFNβ (A), APOBEC3G (B), IRF7 (C), CXCL10 (D), IL-6 (E), IL-10 (F), and IL-12p35 (G) were determined by qPCR (n = 6 donors) (A) and (n = 4 donors) (B–G). Data show the mean values and error bars are the SEM. Statistical analysis was performed using (A–G) two-way ANOVA with Tukey's multiple-comparison test. *$P \leq 0.05$, **$P \leq 0.01$, ***$P \leq 0.001$, ****$P \leq 0.0001$, (A) (n = 6 donors), and (B–G) (n = 4 donors). Source data are available online for this figure.

## Patient consent

To enable comparison between complement and antibody response following infection, we include serum collected in the RECoVERED cohort (van Gils et al, 2022b). Written informed consent was obtained from each study participant. The study design was approved by the local ethics committee of the Amsterdam UMC (Medisch Ethische Toetsingscommissie [METC]; NL73759.018.20). All samples were handled anonymously. Similarly, written informed consent was obtained from all donors of serum samples for complement activation. The Ethics Committee of the Medical University of Innsbruck approved the use of anonymized leftover specimens of COVID-19 patients (ECS1166/2020) and healthy donors (ECS1166/2018) for scientific purposes.

## Reagents and antibodies

The following antibodies were used (all anti-human): CD86 (2331 (FUN-1), BD Pharmingen), CD80 (L307.4, BD Pharmingen), PE-conjugated mouse IgG1 CR3/CD11b (101208, Biolegend), LEAF-purified CR3/CD11b mouse IgG1, LEAF-purified CR4/CD11c mouse IgG1, CR3/CD11b (M1/70), CR4/CD11c (S-HCL-3), CD32 (FUN-2), DC-SIGN (FAB161F, R&D systems), CD16, CD32, CD64 (all BD Pharmingen) and, viability dye (Ghost DyeTM Violet 510, Tonbo Biosciences, San Diego, USA). For extracellular staining, cells were incubated in 0.5% PBS-BSA (Sigma-Aldrich) containing antibodies for 30 min at 4 °C. Single-cell measurements were performed on a Canto flow cytometer II (BD Biosciences) and data was analyzed using FlowJo V10.8.1 (Software by TreeStar). Neutralizing COVA1-18 and non-neutralizing COVA1-27 monoclonal antibodies (mAb) were isolated from participants in the "COVID-19 Specific Antibodies" (COSCA) study (Brouwer et al, 2020) and were generated by Karlijn van der Straten as described previously (Brouwer et al, 2020). Pre-COVID-19 pandemic normal human serum (NHS) generated from a pool of 10 healthy individuals was used in a ratio 1:10. To investigate the effect of mannan on complement activation, pre-COVID-19 NHS was incubated with 100 µg/mL mannan at 37 °C for 1 h. A pool of NHS from a total of 10 healthy individuals was heat-inactivated at 56 °C for 1 h. C3-depleted sera (C3⁻/⁻) was purchased from Sigma (234403) and C3b antibody from Immunotools (21157033).

## Cell lines

The simian kidney cell line VeroE6 (ATCC® CRL-1586™) was cultured in CO₂ independent medium (Gibco Life Technologies, Gaithersburg, MD) supplemented with 10% fetal calf serum (FCS), L-glutamine, and penicillin/streptomycin (10 µg/mL). Cultures were maintained at 37 °C without CO₂. The human embryonic kidney

293T/17 cells (ATCC, CRL-11268) were maintained in Dulbecco's modified Eagle's medium (Gibco Life Technologies) containing 10% FCS, L-glutamine, and penicillin/streptomycin (10 µg/mL).

## DC generation

Human CD14⁺ monocytes were isolated from the blood of healthy volunteer donors (Sanquin blood bank) and subsequently differentiated into monocyte-derived DCs. In short, the isolation from buffy coats was performed by density gradient centrifugation on Lymphoprep (Nycomed) and Percoll (Pharmacia). After Percoll separation, the isolated CD14⁺ monocytes were differentiated into monocyte-derived DCs within 5 days and cultured in RPMI1640 medium (Gibco Life Technologies, Gaithersburg, MD) containing 10% FCS, penicillin/streptomycin (10 µg/mL) and supplemented with the cytokines IL-4 (500 U/mL) and GMCSF (800 U/mL) (both Gibco) (Mesman Annelies et al, 2014). After 4 days of differentiation, DCs were seeded at $1 \times 10^6$/mL in a 96-well plate (Greiner), and after 2 days of recovery, DCs were stimulated or infected as described below. This study was performed in accordance with the ethical principles set out in the Declaration of Helsinki and was approved by the institutional review board of the Amsterdam University Medical Centers, location AMC Medical Ethics Committee and the Ethics Advisory Body of Sanquin Blood Supply Foundation (Amsterdam, Netherlands).

## SARS-CoV-2 pseudovirus production

For the production of single-round infection viruses, human embryonic kidney 293T/17 cells (ATCC, CRL-11268) were co-transfected with an adjusted HIV backbone plasmid (pNL4-3.Luc.R-S-) containing previously described stabilizing mutations in the capsid protein (PMID: 12547912) (Kootstra et al, 2003) and firefly luciferase in the *nef* open reading frame (1.35 µg) and pSARS-CoV-2 expressing SARS-CoV-2 S protein (0.6 µg) (Gen-Bank; MN908947.3), a gift from Paul Bieniasz (Brouwer et al, 2020; Schmidt et al, 2020). Transfection was performed in 293T/17 cells using genejuice (Novagen, USA) transfection kit according to the manufacturer's protocol. At day 3 or day 4, pseudotyped SARS-CoV-2 virus particles were harvested and filtered over a 0.45-µm nitrocellulose membrane (Sartorius Stedim, Gottingen, Germany). SARS-CoV-2 pseudovirus productions were quantified by RETRO-TEK HIV-1 p24 ELISA according to manufacturer instructions (ZeptoMetrix Corporation).

## SARS-CoV-2 isolate (hCoV-19/Italy-WT)

The wild-type (WT) authentic SARS-CoV-2 virus hCoV-19/WT (D614G variant) was obtained from Dr. Maria R. Capobianchi

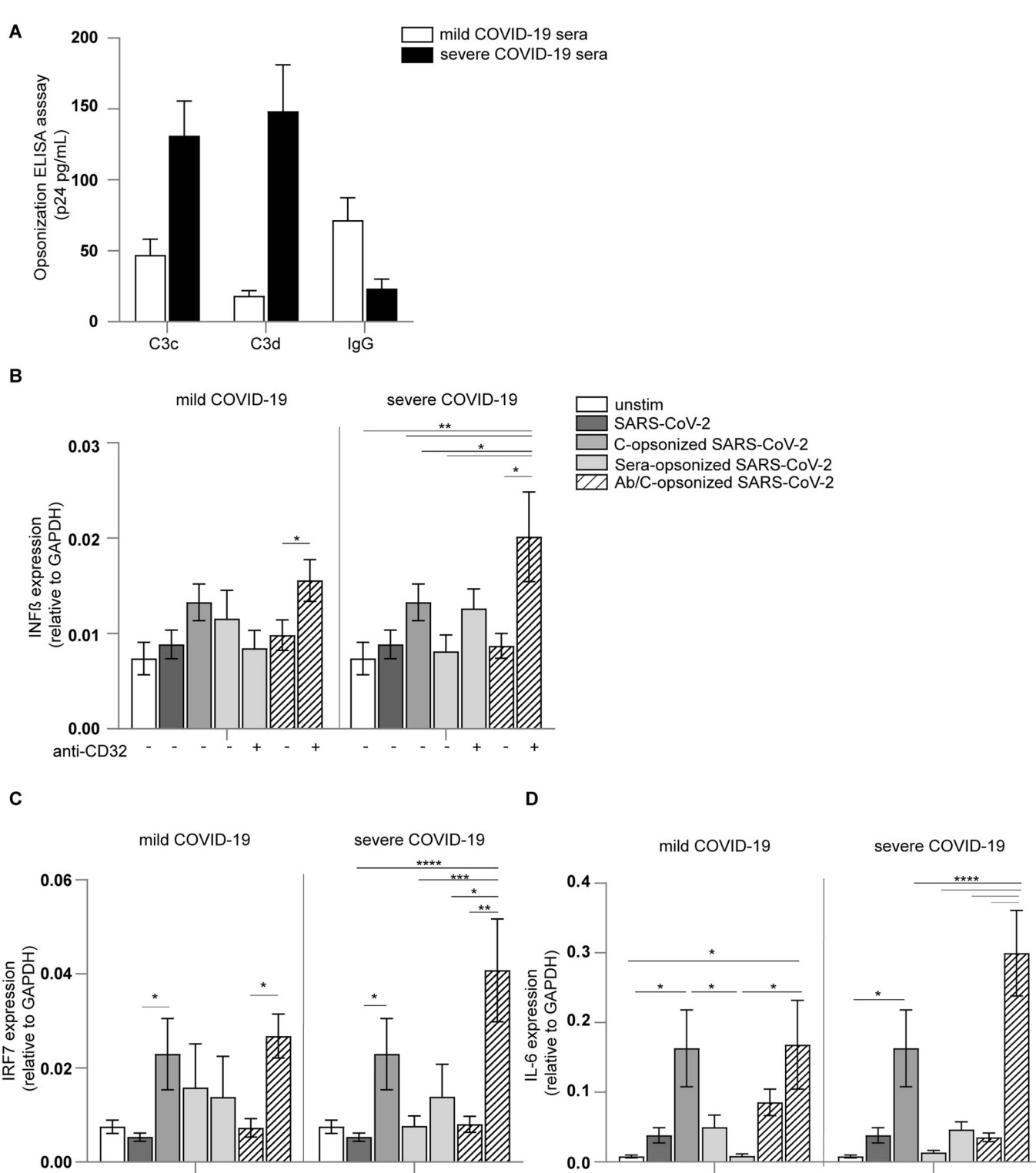

through BEI Resources, NIAID, NIH: SARS-Related Coronavirus 2, Isolate Italy-INMI1, NR-52284, originally isolated on January 2020 in Rome, Italy. SARS-CoV-2 authentic virus stocks from primary isolates were generated in VeroE6 cells. Cytopathic effect (CPE) formation was monitored and after 48 h the virus supernatant was harvested. Viral titers were determined by tissue-cultured infectious dose (TCID50) on

VeroE6 cells. Briefly, VeroE6 cells were seeded in a 96-well plate at a cell density of 10,000 cells/well. The following day, cells were inoculated with a five-fold serial dilution of SARS-CoV-2 isolate in quadruplicate. Cell cytotoxicity was measured by using an MTT assay 48 h post infection. Loss of MTT staining, analyzed by a spectrometer ($OD_{580nm}$) was indicative of SARS-CoV-2-induced CPE. Viral titer

**Figure 6. Disease severity dictates SARS-CoV-2 complement activation and antibody response.**

(A) SARS-CoV-2 pseudovirus opsonization patterns with mild and severe COVID-19 patient sera was determined by ELISA (p24 pg/mL) using anti-human C3c and C3d, for iC3b recognition, and anti-human IgG, for immunoglobulins detection (n = 4 donors). (B) Human monocyte-derived DCs were exposed to SARS-CoV-2 isolate (hCoV-19/Italy-WT, 1000 TCID/mL), to complement-opsonized SARS-CoV-2 (hCoV-19/Italy-WT, 1000 TCID/mL), COVID-19 patient serum (mild or severe) and antibody/complement-opsonized SARS-CoV-2 (hCoV-19/Italy-WT, 1000 TCID/mL) in presence or absence of anti-CD32 for 2 h and 6 h. mRNA levels of IFNβ (B) were determined after 2 h and mRNA levels of IRF7 (C) and IL-6 (D) after 6 h by qPCR (n = 8 donors) (C, D). Data show the mean values and error bars are the SEM. Statistical analysis was performed using (B–D) two-way ANOVA with Sidak's multiple-comparison test. *$P \le 0.05$, **$P \le 0.01$, ***$P \le 0.001$, ****$P \le 0.0001$ (n = 8 donors). Source data are available online for this figure.

was determined as TCID50/mL and calculated based on the method first proposed by Reed and Muench (Reed and Muench, 1938). All experiments with the WT SARS-CoV-2 isolate (hCoV-19/Italy-WT) were performed in a BSL-3 laboratory, following appropriate safety and security protocols approved by the Amsterdam UMC BioSafety-Groep and performed under the environmental license obtained from the municipality of Amsterdam.

## SARS-CoV-2 (hCoV-19/Italy-WT) neutralization assay

Antibody neutralization activity of SARS-CoV-2 infection was tested as previously described (Caniels et al, 2021b) and included some modifications. Briefly, VeroE6 cells were seeded at a density of 10,000 cells/well in a 96-well plate one day prior to the start of the neutralization assay. Heat-inactivated sera samples were serially diluted in cell culture medium $CO_2$ independent medium (Gibco life Technologies, Gaithersburg, MD) supplemented with 10% fetal calf serum (FCS), L-glutamine and penicillin/streptomycin (10 µg/mL), mixed 1:1 ratio with authentic SARS-CoV-2 and incubated for 1 h at 37 °C. Subsequently, these mixtures were added to the cells in a 1:1 ratio and incubated for 48 h at 37 °C without $CO_2$, followed by an MTT assay. Loss of MTT staining, analyzed by a spectrometer ($OD_{580nm}$) was indicative of SARS-CoV-2-induced CPE. The neutralization titers (IC50) were determined as the serum dilution or antibody concentration at which infectivity was inhibited by 50%, respectively, using a non-linear regression curve fit (GraphPad Prism software version 8.3) and serum dilutions were converted into international units per mL (IU/mL) using the WHO International Standard for anti-SARS-CoV-2 immunoglobulin (NIBSC 20/136).

## Opsonization assay of SARS-CoV-2 pseudovirus and hCoV-19/Italy-WT

Incubation of SARS-CoV-2 with pre-COVID-19 pandemic pooled normal human serum (NHS) mediated covalent deposition of C3 fragments (C3b, iC3b, C3d, C3c) and specific IgGs on the viral surfaces. To mimic the in vivo situation (Jarlhelt et al, 2021), where SARS-CoV-2 is opsonized with complement or IgGs, SARS-CoV-2 pseudovirus (191.05 ng/mL of SARS-CoV-2 pseudovirus) and authentic SARS-CoV-2 (hCoV-19/Italy-WT, 1000 TCID/mL) were incubated for 1 h at 37 °C with pre-COVID-19 pandemic NHS (1:10 ratio), as complement source to obtain complement-opsonized SARS-CoV-2; with specific IgGs or highly IgG content sera, to obtains IgG-opsonized SARS-CoV-2, or a combination of both (C-Ig) in a 1:10 ratio (Sullivan et al, 1996). As a negative control, the virus was incubated under the same conditions with plain RPMI1640. Serum from at least 10 healthy donors, referred to as NHS, were pooled and stored at −80 °C. The presence of C3 fragments and IgGs on the viral surface was detected by

opsonization ELISA assay, also called Viral Capture Assay (VCA) as previously described (Frank et al, 1996; Nijmeijer et al, 2021; Wilflingseder et al, 2007). Briefly, 96-well plates were coated with rabbit anti-mouse IgG (DAKO) at 4 °C overnight. ELISA plates were coated anti-human C3c and C3d as well as human IgG and incubated overnight with differentially opsonized virus preparations (1 ng/p24 per well) at 4 °C and extensively washed with RPMI1640 medium to remove unbound virus. Mouse IgG antibodies were used as control for background binding. For the SARS-CoV-2 pseudovirus, viral samples were lysed (1% Triton) and binding was quantified by p24 ELISA to confirm the opsonization pattern (Purtscher et al, 1994). The opsonization pattern of SARS-CoV-2 isolate (hCoV-19/Italy-WT) was determined by qPCR. The viral samples were lysed, and SARS-CoV-2 RNA was extracted using FavorPrep Viral RNA Minikit (FAVORGEN, Ping-Tung, Taiwan), according to the manufacturer's instructions. Sequences specific to region N1 of the Nucleocapsid gene published on the CDC website (https://www.cdc.gov/coronavirus/2019-ncov/lab/rt-pcr-panel-primer-probes.html) were used. Luna Universal Probe One-Step RT-PCR kit (New England BioLabs, Ipswich, Mass) was used for target amplification, and runs were performed on the CFX96 real-time detection system (Bio-Rad). For absolute quantification using the standard curve method, SARS-CoV-2 RNA was obtained as a PCR standard control from the National Institute for Biological Standards and Control UK (Ridge, UK).

## COVID-19 patient serum

Pooled serum from at least 20 random individuals (mild/moderate disease) all in 2020 (Wuhan variant, no vaccination), collected 19 days post-symptom onset with high neutralizing antibody content, was heat-inactivated (1 h at 56 °C) to destroy complement activity. In brief, SARS-CoV-2 isolate (hCoV-19/Italy-WT, 1000 TCID/mL) was incubated for 1 h at 37 °C with high neutralizing antibody pooled serum in a 1:10 ratio, to generate antibody-opsonized SARS-CoV-2.

Serum from ten COVID-19 patients after natural infection, with either mild (n = 5) or severe (n = 5) disease outcome, collected ~3 months post infection, were used in a 1:10 ratio as antibody-mediated-complement opsonization source for opsonization and generate complement- and antibody-opsonized SARS-CoV-2. The ability to opsonize the virions was assessed by ELISA, as previously described (Frank et al, 1996; Nijmeijer et al, 2021).

Measurements of plasma anaphylatoxin were carried out as indicator of in vivo complement activation in COVID-19 patients with mild (n = 7) and severe (n = 7) disease progression were included after 18–61 days of positive SARS-CoV-2 PCR. In addition, sera prior PBMCs isolation from seven healthy donors collected before the COVID-19 pandemic (before October 2019) were used as negative controls.

## Virus binding and internalization

In order to determine SARS-CoV-2 binding and internalization, target cells were seeded in a 96-well plate at a density of 100,000 cells in 100 µl. Cells were exposed to SARS-CoV-2 isolate (hCoV-19/Italy-WT, 1000 TCID/mL) for 4 h at 4 °C for binding. After 4 h of incubation, cells were washed extensively to remove the unbound virus. Cells were lysed with AVL buffer and RNA was isolated with the QIAmp Viral RNA Mini Kit (Qiagen) according to the manufacturer's protocol.

## DC stimulation and infection

DCs were left unstimulated or stimulated with 10 ng/mL LPS *Salmonella typhosa* (Sigma) and SARS-CoV-2 isolate (hCoV-19/Italy-WT) with different opsonization patterns SARS-CoV-2, C-opsonized SARS-CoV-2, Ab-opsonized SARS-CoV-2 and Ab/C-opsonized SARS-CoV-2 at 1000 TCID/mL. Blocking of CR3/CD11b (LEAF-purified CR3/CD11b) and CR4/CD11c (LEAF-purified CR4/CD11c) was performed with 10 µg/mL for 30 min at 37 °C before adding the virus preparations. Similarly, for blocking of FcγRII, DCs were treated with anti-CD32 antibody 1 µg/mL for 1 h at 37 °C. DCs do not express ACE2 and are therefore not infected by SARS-CoV-2 (van der Donk et al, 2022b). Therefore, viral stimulation SARS-CoV-2 isolate (hCoV-19/Italy-WT) at 1000 TCID/mL (MOI 0.028) was performed for 2 h, 6 h after which the cells were lysed for RNA isolation and cytokine production analysis. In addition, cells were stimulated for 24 h and fixed for 30 min with 4% paraformaldehyde to assess the maturation phenotype with flow cytometry.

Viral infection and secretion were determined by RT-PCR measurement of ORF-1b transcript. DCs were exposed to SARS-CoV-2 isolate (hCoV-19/Italy, 1000 TCID/ml) and after 24 h at 37 °C, cells were extensively washed to remove unbound virus. Following 24 h in clean culture media, cell supernatant was taken for isolation of viral RNA to investigate productive infection. DCs were incubated with SARS-CoV-2 isolate (hCoV-19/Italy, 1000 TCID/mL) for 16 h and 24 h after which ORF-1b transcript was determined by RT-PCR. In parallel, HuH7.5 cells were infected with SARS-CoV-2 isolate (hCoV-19/Italy, 1000 TCID/mL) for 24 h at 37 °C. Subsequently, Huh7.5 cells were lysed for isolation of viral RNA, as control for direct infection.

## Caspase-1 activity

Active caspase-1 was detected using the FAM-FLICA Caspase-1 Assay kit (Immunochemistry Technologies) according to the manufacturer's instructions. In brief, DCs were washed in IMDM medium lacking phenol red (Gibco), supplemented with 10% FCS, penicillin, and streptomycin (100 U/mL and 100 µg/mL, respectively, Thermo Fisher) prior to stimulations. After 14 h, DCs were treated with FAM-FLICA reagent and incubated at 37 °C, 5% $CO_2$ for 1 h. Cells were washed three times in apoptotic wash buffer (Immunochemistry Technologies), immediately followed by flow cytometry analysis using the FACSCanto II (BD Bioscience) and FlowJo software version 10.7, and guidelines for the use of flow cytometry and cell sorting in immunological studies were followed. Live cells were gated based on FSC/SSC and the caspase-1[+] (FAM-FLICA) population within this live-cell population was assessed.

## RNA isolation and quantitative real-time PCR

Cells incubated with SARS-CoV-2 isolate (hCoV-19/Italy-WT) were lysed with AVL buffer, and RNA was isolated with QIAmp Viral RNA Mini Kit (Qiagen) according to the manufacturer's protocol. cDNA was synthesized with M-MLV reverse transcriptase Kit (Promega) and diluted 1/5 before further application. PCR amplification was performed by using RT-PCR in the presence of SYBR green in a 7500 Fast Real-time PCR System (ABI). Specific primers were designed with Primer Express 2.0 (Applied Biosystems). The following primer sequences were used:

GAPDH: F-primer 5′-CCATGTTCGTCATGGGTGTG-3′, R-primer 5′-GGTGCTAA GCAGTTGGTGGTG-3′; SARS-CoV-2 ORF-1b: F-primer 5′-TGGGGTTTTACAGGTAACCT-3′, R-primer 5′-AACACG CTTAACAAAGCACTC-3′; IFNB: F-primer 5′-ACAGACTTACAGG TTACCTCCGAAAC-3′, R-primer 5′-CATCTGCTGGTTGAAGAA TGCTT-3′; CXCL10: F-primer 5′-CGCTGTACCTGCATCAGCAT-3′; R-primer 5′-CATCTCTTCTCACCCTTCTTTTTCA-3′ ; IL-6: F-primer 5′-TGCAATAACCACCCCTGACC-3′, R-primer 5′-TGCGCAGAATG AGATGAGTTG-3′; IL-10: F-primer 5′-GAGGCTACGGCGCTGTCAT-3′, R-primer 5′-CCACGGCCTTGCTCTTGTT-3′; IL-12p35: F-primer TGGACCACCTCAGTTTGGC; R-primer TTCCTGGGTCTGGA GTGGC.

The normalized amount of target mRNA was calculated from the Ct values obtained for both target and household mRNA with the equation $Nt = 2^{Ct\,(GAPDH)\,-\,Ct(target)}$.

## ELISA

Cell supernatants were harvested after 24 h of stimulation, and secretion of IL-1β and CXCL10 were measured by ELISA (eBiosciences and BD Biosciences, respectively) according to the manufacturer's instructions. The supernatant containing SARS-CoV-2 was inactivated with 1% triton before performing the ELISA. $OD_{450nm}$ values were measured using BioTek Synergy HT.

C3a and C5a levels of plasma samples from COVID-19 patients and healthy donors were detected by the BD OptEIA Human C3a ELISA Kit and BD OptEIA Human C5a ELISA kit, respectively (BD Biosciences, Franklin Lakes, NJ, USA), according to the manufacturer's instructions.

## Statistics

All results are presented as mean ± SEM and were analyzed by GraphPad Prism 9 software (GraphPad Software Inc.). A two-tailed, parametric Student's *t* test for paired observations (differences within the same donor) or unpaired observation, Mann–Whitney tests (differences between different donors that were not normally distributed) was performed. For unpaired, non-parametric observations a one-way ANOVA or two-way ANOVA test with post hoc analysis (Tukey's, Sidak 's or Dunnett's) was performed. Statistical significance was set at $*P < 0.05$; $**P < 0.01$; $***P < 0.001$; $****P < 0.0001$.

# Data availability

This study includes no data deposited in external repositories.

## Peer review information

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

## Acknowledgements

The authors thank Jonne Snitselaar, Yoann Aldon and Judith Burger for help with the production of antibodies and pseudovirus reagents. In addition, the authors thank Elke Wynberg and Hugo DG van Willigen for their contribution to The *RECoVERED* Study This research was funded by the Netherlands Organisation for Health Research and Development (ZonMw) together with the Stichting Proefdiervrij (ZonMw MKMD COVID-19 grant nr.114025008 to TBHG), and European Research Council (Advanced grant 670424 to TBHG), Amsterdam UMC PhD grant and two COVID-19 grants from the Amsterdam Institute for Infection & Immunity (to TBHG, RWS, and MJG). LEHvdD was supported by the Netherlands Organization for Scientific Research (NWO) (Grant No. 91717305). MBJ and this research were supported by a Work Visit Grant of the Amsterdam Institute for Infection and Immunity and by the APART-MINT Fellowship of the Austrian Academy of Sciences (ÖAW- APART-MINT Grant No. 11978).

## Author contributions

**Marta Bermejo-Jambrina**: Conceptualization; Data curation; Formal analysis; Supervision; Funding acquisition; Validation; Investigation; Methodology; Writing—original draft; Project administration; Writing—review and editing. **Lieve EH van der Donk**: Conceptualization; Data curation; Formal analysis; Investigation; Methodology; Writing—original draft. **John L van Hamme**: Data curation; Investigation; Methodology. **Doris Wilflingseder**: Conceptualization; Resources; Supervision. **Godelieve de Bree**: Resources. **Maria Prins**: Resources. **Menno de Jong**: Resources. **Pythia Nieuwkerk**: Resources. **Marit J van Gils**: Resources; Data curation; Writing—review and editing. **Neeltje A Kootstra**: Resources; Data curation; Validation; Investigation; Methodology; Writing—review and editing. **Teunis BH Geijtenbeek**: Conceptualization; Resources; Formal analysis; Supervision; Funding acquisition; Validation; Writing—original draft; Project administration; Writing—review and editing.

## Disclosure and competing interests statement

The authors declare no competing interests.

# Expanded View Figures

**Figure EV1.  Spike-dependent opsonization requires C3 deposition for complement opsonization.**

(**A**) Particles lacking SARS-CoV-2 Spike glycoprotein and SARS-CoV-2 pseudovirus opsonisation were determined by ELISA (p24 pg/mL) ($n = 3$ donors). (**B**) Deposition of C3b on DC-internalized SARS-CoV-2 after incubation with pre-pandemic NHS, C3-depleted sera and heat-inactivated sera ($n = 3$ donors). (**C**, **D**) Human monocyte-derived DCs were exposed to SARS-CoV-2 isolate (hCoV-19/Italy-WT, 1000 TCID/mL) and complement-opsonized SARS-CoV-2 (hCoV-19/Italy-WT, 1000 TCID/mL) in presence or absence of anti-CD11b and anti-CD11c. LPS stimulation was used as positive control for DC maturation, which was measured after 24 h by flow cytometry. Cumulative flow cytometry data of CD11b and CD11c ($n = 12$ donors). Data show the mean values and error bars are the SEM. Statistical analysis was performed using (**A**) two-way ANOVA with Tukey multiple-comparison test. *$P \leq 0.05$, **$P \leq 0.01$, ***$P \leq 0.001$ ($n = 3$ donors). (**B**) ordinary one-way ANOVA with Dunnett's multiple-comparison test **$P \leq 0.01$ ($n = 3$ donors). (**C**, **D**) Two-way ANOVA with Tukey multiple-comparison test. *$P \leq 0.05$, ***$P \leq 0.001$ ($n = 12$ donors). Source data are available online for this figure.

**A**

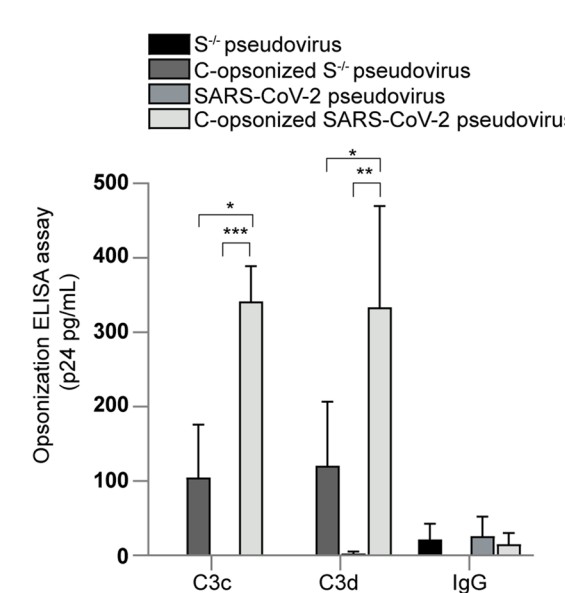

**B**

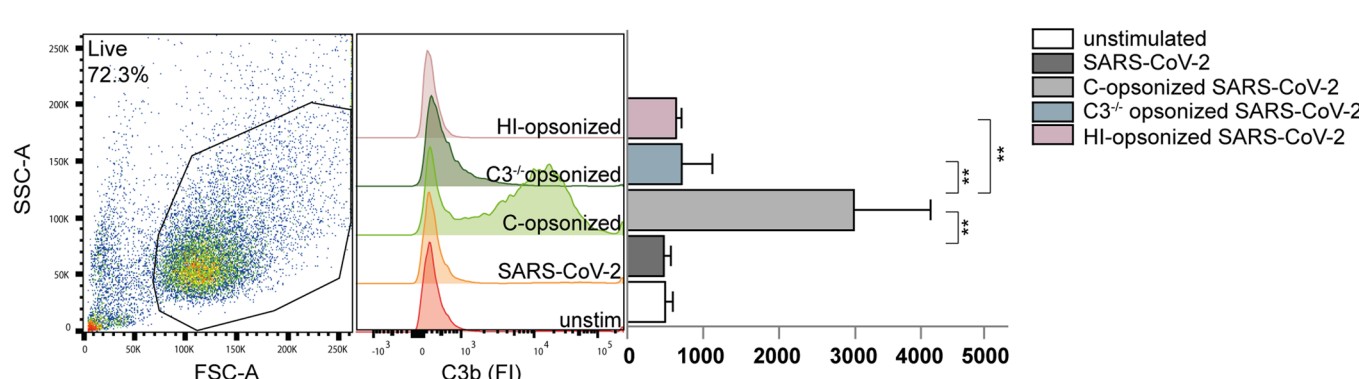

**C**

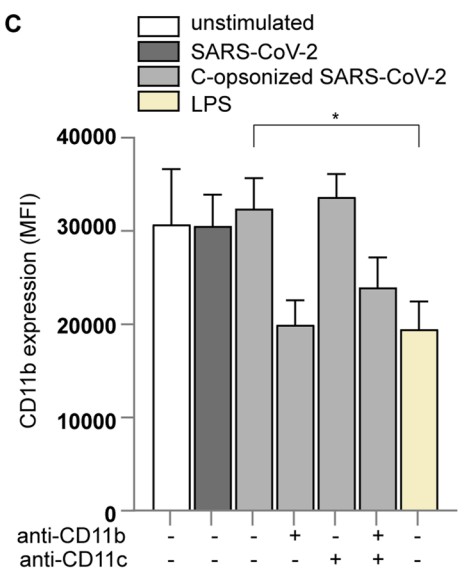

**D**

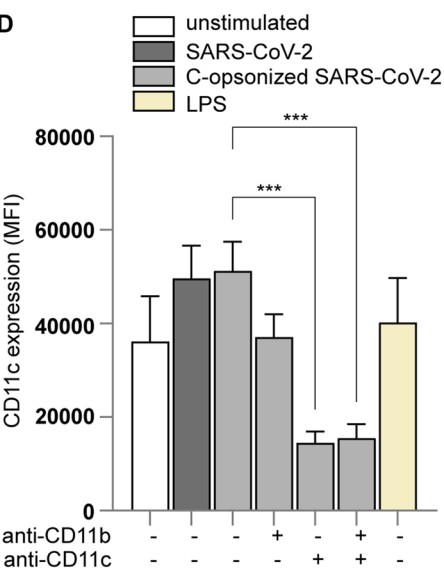

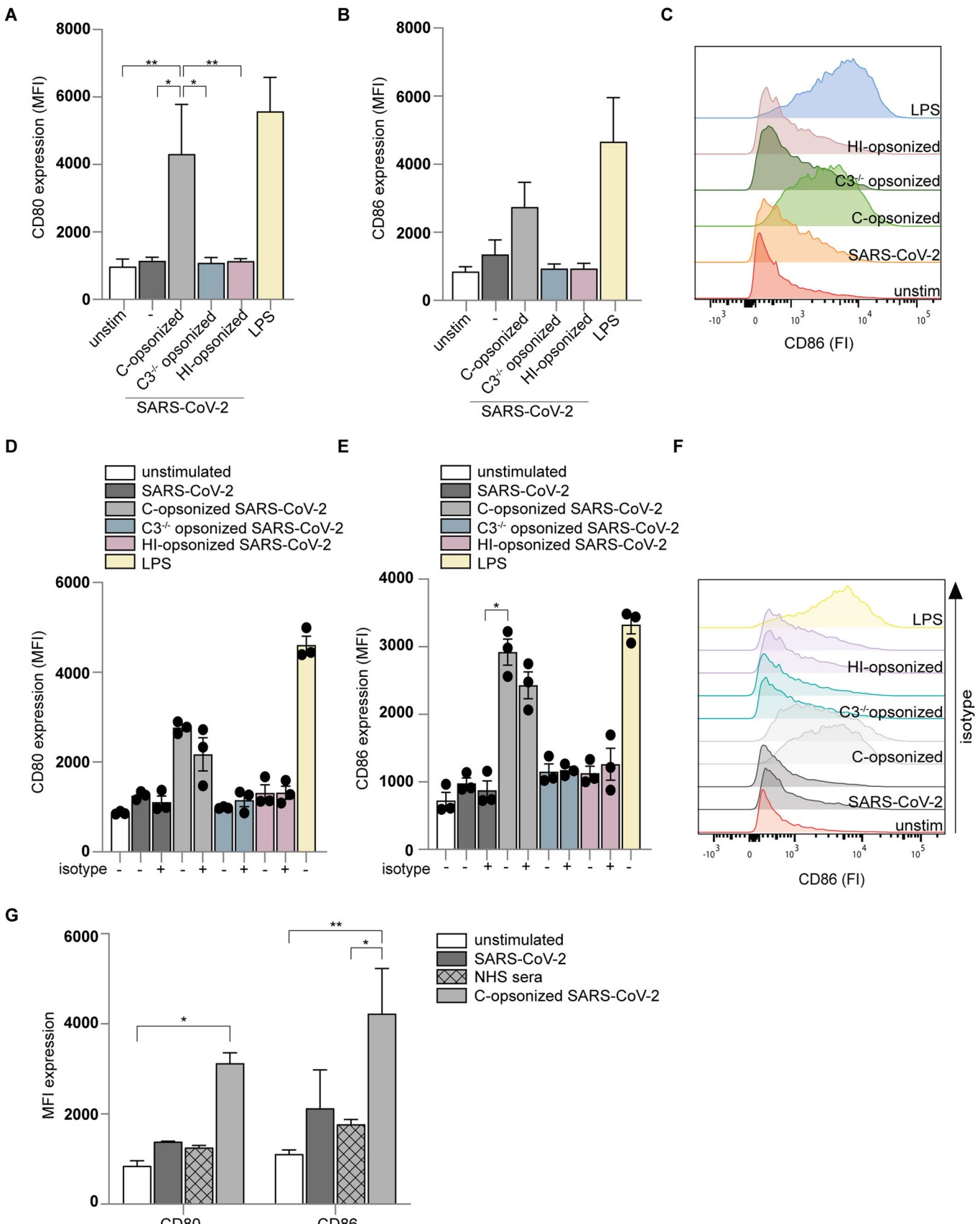

◀ **Figure EV2. C3 deposition is required for DC maturation.**

(**A–F**) Human monocyte-derived DCs were exposed to SARS-CoV-2 isolate (hCoV-19/Italy-WT, 1000 TCID/mL), complement-opsonized SARS-CoV-2 (hCoV-19/Italy-WT, 1000 TCID/mL), C3-depleted-opsonized SARS-CoV-2 (hCoV-19/Italy-WT, 1000 TCID/mL) and heat-inactivated-opsonized SARS-CoV-2 (hCoV-19/Italy-WT, 1000 TCID/mL) in absence (**A–C**) or presence (**D–F**) of an isotype. LPS stimulation was used as positive control for DC maturation, which was measured after 24 h by flow cytometry. Cumulative flow cytometry data of CD80 (**A–D**) and CD86 (**B–E**) (*n* = 5 donors). (**C, D**) Representative histograms of CD86 expression. (**G**) DCs were exposed to SARS-CoV-2 isolate (hCoV-19/Italy-WT, 1000 TCID/mL), pre-pandemic NHS sera and complement-opsonized SARS-CoV-2 (hCoV-19/Italy-WT, 1000 TCID/mL) and the expression of CD80 and CD86 markers were measured (*n* = 4 donors). Data show the mean values and error bars are the SEM. Statistical analysis was performed using (**A**) ordinary one-way ANOVA with Dunnett's multiple-comparison test. *$P \leq 0.05$, **$P \leq 0.01$ (*n* = 5 donors). (**F**) ordinary one-way ANOVA with Tukey multiple-comparison test. *$P \leq 0.05$ (*n* = 5 donors). (**G**) ordinary one-way ANOVA with Tukey multiple-comparison test. *$P \leq 0.05$, **$P \leq 0.01$ (*n* = 4 donors). Source data are available online for this figure.

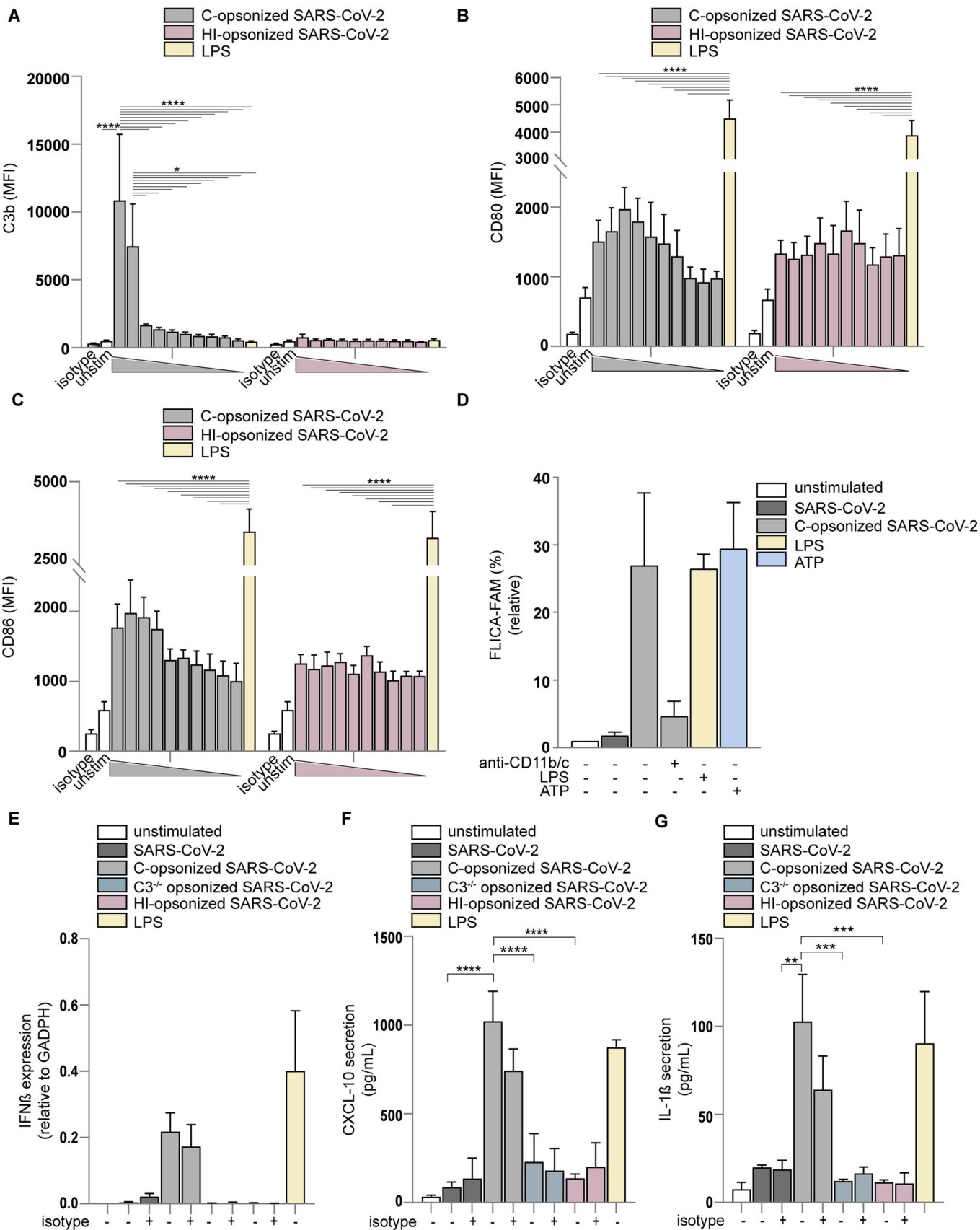

◄ **Figure EV3.  SARS-CoV-2 opsonization is concentration-dependent and requires C3.**

(A) Mean fluorescence index (MFI) of serum concentration-dependent C3b deposition on DC-internalized complement-opsonized SARS-CoV-2 and heat-inactivated-opsonized SARS-CoV-2 after 24 h ($n = 4$ donors). (B, C) Mean fluorescence index (MFI) of serum concentration-dependent co-stimulatory markers, CD80 (B) and CD86 (C) on DC activated cells after exposure to complement-opsonized SARS-CoV-2 and heat-inactivated-opsonized SARS-CoV-2 after 24 h ($n = 4$ donors). LPS stimulation was used as positive control for DC maturation, which was measured after 24 h by flow cytometry. (D) Percentages of FLICA$^+$ from different stimulated DC ($n = 3$ donors). (E–G) Human monocyte-derived DCs were exposed to SARS-CoV-2 isolate (hCoV-19/Italy-WT, 1000 TCID/mL), complement-opsonized SARS-CoV-2 (hCoV-19/Italy-WT, 1000 TCID/mL), C3-depleted-opsonized SARS-CoV-2 (hCoV-19/Italy-WT, 1000 TCID/mL) and heat-inactivated-opsonized SARS-CoV-2 (hCoV-19/Italy-WT, 1000 TCID/mL), as well as LPS (100 ng/mL) in presence or absence of an isotype for 2 h (E) and 24 h (F, G). mRNA levels of IFNβ after 2 h were measured by qPCR ($n = 7$ donors) (E). CXCL10 and IL-1β secretion (pg/mL) in the supernatant were measured after 24 h by ELISA ($n = 4$ donors). Data show the mean values and error bars are the SEM. Statistical analysis was performed using (A–C), two-way ANOVA with Tukey multiple-comparison test. *$P ≤ 0.05$, ****$P ≤ 0.0001$ ($n = 4$ donors) (A–C). (E–G) Two-way ANOVA with Tukey multiple-comparison test. **$P ≤ 0.01$, ***$P ≤ 0.001$, ****$P ≤ 0.0001$ ($n = 7$ donors) (E) and ($n = 4$ donors) (F, G). Source data are available online for this figure.

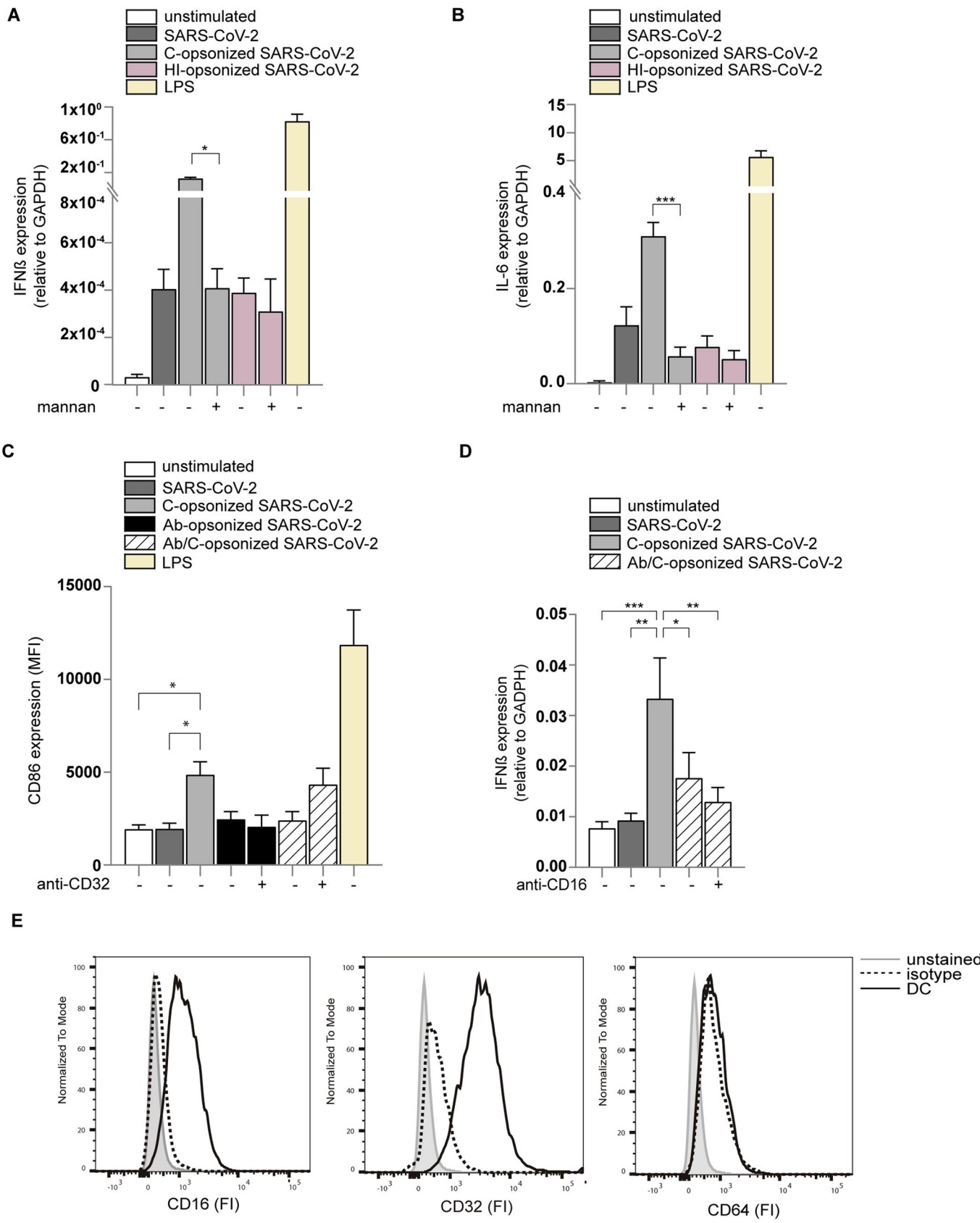

◀  **Figure EV4.  Complement-mediated DC activation and antiviral response is dependent on lectin pathway.**

(**A, B**) NHS and HIS were incubated with mannan (100 µg/mL), prior SARS-CoV-2 opsonization. DCs were exposed to non-, complement-opsonized SARS-CoV-2 and heat-inactivated-opsonized SARS-CoV-2 in presence or absence of mannan, and mRNA levels of IFNβ after 2 h (*n* = 3 donors) (**A**) and IL-6 after 6 h (*n* = 3 donors) (**B**) were determined by qPCR. (**C**) Human monocyte-derived DCs were exposed to SARS-CoV-2 isolate (hCoV-19/Italy-WT, 1000 TCID/mL), to complement-opsonized SARS-CoV-2 (hCoV-19/Italy-WT, 1000 TCID/mL), to antibody-opsonized SARS-CoV-2 (hCoV-19/Italy-WT, 1000 TCID/mL) and to antibody/complement-opsonized SARS-CoV-2 (hCoV-19/Italy-WT, 1000 TCID/mL) in presence or absence of anti-CD32 for 24 h. LPS stimulation was used as positive control for DC maturation, which was measured after 24 h by flow cytometry. Cumulative flow cytometry data of CD86 (*n* = 12 donors). (**D**) Human monocyte-derived DCs were exposed to SARS-CoV-2 isolate (hCoV-19/Italy-WT, 1000 TCID/mL), to complement-opsonized SARS-CoV-2 (hCoV-19/Italy-WT, 1000 TCID/mL) and to antibody/complement- opsonized SARS-CoV-2 (hCoV-19/Italy-WT, 1000 TCID/mL) in presence or absence of anti-CD16 for 2 h, and mRNA levels of IFNβ (*n* = 6 donors) were determined by qPCR. (**E**) DCs were stained with antibodies against the surface markers CD16, CD32 and CD64 and analyzed by flow cytometry. Representative histograms for an experiment repeated more than three times with similar results (*n* = 3 donors). Data show the mean values and error bars are the SEM. Statistical analysis was performed using (**A, B**) two-way ANOVA with Tukey multiple-comparison test. *$P \leq 0.05$, ***$P \leq 0.001$ (*n* = 3 donors). (**C**) ordinary one-way ANOVA with Tukey multiple-comparison test. *$P \leq 0.05$ (*n* = 6 donors). (**D**) Two-way ANOVA with Tukey's multiple-comparison test. *$P \leq 0.05$, **$P \leq 0.01$, ***$P \leq 0.001$ (*n* = 6 donors). Source data are available online for this figure.

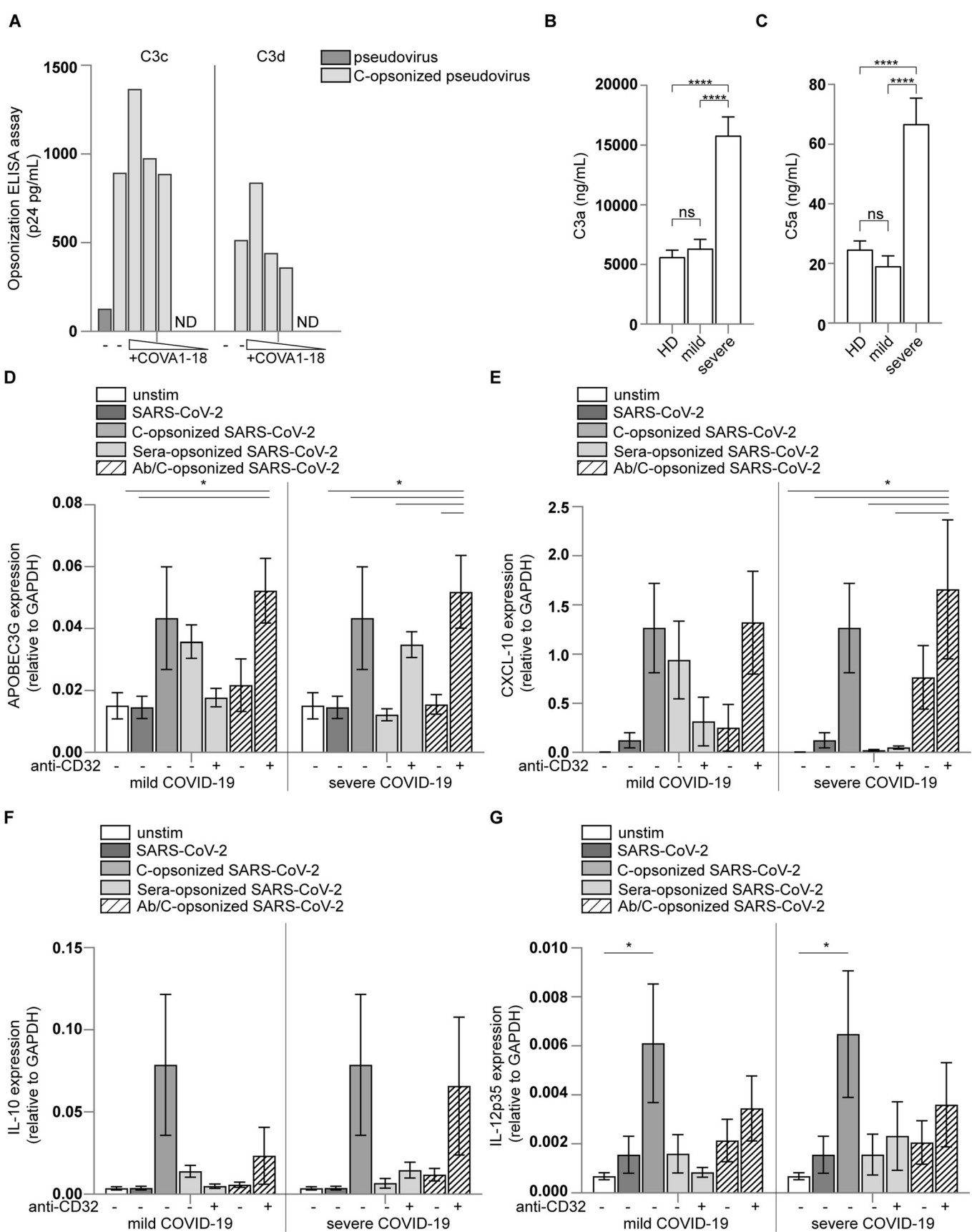

◀　**Figure EV5.　Increased complement activation is a distinctive feature of severe COVID-19 patients.**

(A) NHS concentration-dependent SARS-CoV-2 pseudovirus opsonization by C3c and C3d were determined by (A) ELISA (p24 pg/mL) ($n = 2$ donors) (B, C) C3a and C5a level were determined in healthy donors, mild and severe COVID-19 patients ($n = 7$ donors per group). Plasma samples were harvested and C3a (ng/mL) and C5a (ng/mL) levels were analyzed using a BD Biosciences OptEIA Human C3a and C5a ELISA kit. (D–G) Human monocyte-derived DCs were exposed to SARS-CoV-2 isolate (hCoV-19/Italy-WT, 1000 TCID/mL), to complement-opsonized SARS-CoV-2 (hCoV-19/Italy-WT, 1000 TCID/mL), COVID-19 patient serum (mild or severe) and antibody/complement-opsonized SARS-CoV-2 (hCoV-19/Italy-WT, 1000 TCID/mL) in presence or absence of anti-CD32 for 6 h. mRNA levels of APOBEC3G (D) CXCL10 (E), IL-10 (F) and IL-12p35 (G) after 6 h were determined by qPCR ($n = 8$ donors). Data show the mean values and error bars are the SEM. Statistical analysis was performed using (B, C) ordinary one-way ANOVA with Tukey multiple-comparison test. ****$P \leq 0.0001$ ($n = 7$ donors). (D, E, G) Two-way ANOVA with Tukey's multiple-comparison test. *$P \leq 0.05$ ($n = 8$ donors). Source data are available online for this figure.

　　　　　　　　　　　　　　　　　　　　