## [Peer Review File · The EMBO Journal]

Control of complement-induced inflammatory responses to SARS-CoV-2 infection by anti-SARS-CoV-2 antibodies

Marta Bermejo-Jambrina, Lieve van der Donk, John van Hamme, Doris Wilflingseder, Godelieve de Bree, Maria Prins, Menno de Jong, Pythia Nieuwkerk, Marit van Gils, Neeltje Kootstra, and Teunis Geijtenbeek

Corresponding author(s): Teunis Geijtenbeek (t.b.geijtenbeek@amsterdamumc.nl) , Marta Bermejo-Jambrina (m.bermejojambрина@amsterdamumc.nl)

Review Timeline:

Submission Date:	9th Jun 23
Editorial Decision:	20th Jul 23
Revision Received:	1st Dec 23
Editorial Decision:	5th Jan 24
Revision Received:	31st Jan 24
Accepted:	6th Feb 24

Editor: Karin Dumstrei / Ioannis Papaioannou

Transaction Report:

Dear Teunis,

Thank you for submitting your manuscript to The EMBO Journal. Your study has now been seen by three referees and their comments are provided below.

As you can see from the comments, the referees find the analysis interesting but also raise some concerns regarding the experimental data supporting the key conclusions. Should you be able to extend the analysis and add more data along the lines suggested below then I would like to invite a revised version.

Let me know if we need to discuss anything further.

best wishes

karin

Karin Dumstrei, PhD
Senior Editor
The EMBO Journal

We realize that it is difficult to revise to a specific deadline. In the interest of protecting the conceptual advance provided by the work, we recommend a revision within 3 months (18th Oct 2023). Please discuss the revision progress ahead of this time with the editor if you require more time to complete the revisions. Use the link below to submit your revision:

Referee #1:

The authors investigated the effect of complement, anti-SARS-CoV-2 antibodies, and cell surface receptors on activation of monocyte-derived DCs by SARS-CoV-2 virus, or by lentiviral vectors pseudotyped with Spike. SARS-CoV-2 association with DCs, and activation of DCs to produce inflammatory cytokines and IL-1, was increased by preincubation of virions with pre-COVID-19 normal human serum (NHS). This activity was attributed to the accumulation of C3c and C3d from the serum on the virus surface. DC activation was blocked by antibodies to the DC cell surface complement receptors CR3 and CR4, and by mannan, an inhibitor of the lectin pathway. The authors then showed that convalescent serum containing anti-SARS-CoV-2 antibodies, as well as anti-Spike monoclonal antibodies, blocked complement-mediated DC activation. Inhibition of DC activation by convalescent sera or by anti-Spike monoclonal antibodies was reversed by anti-FcγRII (CD32) antibody. When serum from people with mild COVID-19 was compared with that from people with severe COVID-19, correlation was shown between disease severity and degree of virion opsonization, indicating that COVID-19 severity correlated with greater complement activation.

Overall, this manuscript describes interesting links between SARS-CoV-2 opsonization by serum complement and innate immune activation of dendritic cells, and correlation between these observations and disease severity. The experiments presented are in themselves reasonably convincing though in most experiments, the authors have derived conclusions based on a single assay. The manuscript would be strengthened if the authors buttressed their findings with some orthogonal experimental approaches and controls. A few suggestions are provided here, along with some minor technical comments and questions:

1. Of the various complement proteins present in sera, why did the authors focus on C3c and C3d for their opsonization experiments. Might other complement components like C3b contribute to opsonization?
2. The authors claim that opsonization and DC activation was due to C3c and C3d present in sera. But sera is a complex mixture of proteins and they only showed that C3c and C3d were associated with the virions. Is there an experiment that can be done to show that C3c and/or C3d are required for opsonization activity and DC activation? For example, does purified C3c and/or C3d opsonize? Alternatively, does depletion of C3c and/or C3d from serum disrupt opsonization?
3. Fig 1A: The authors claim that opsonization requires Spike but they do not have a control to show that opsonization of lentivectors requires Spike. Does complement opsonize lentivectors in the absence of Spike or pseudotyped with other viral glycoproteins? Are the experiments in Fig 1A statistically significant?
4. Fig 1B: Complement seems to increase virus binding to DCs by ~2-fold. This raises a few questions:
 - Is binding in the absence of complement due to heparan sulfate?
 - If so, why do anti-CD11b/c antibodies decrease binding to levels that are lower than seen in the absence of complement?
 - What would be the effect of non-specific antibody or of antibody against another DC cell surface molecule?
5. Fig 1D and G: There seems to be a discrepancy between panels D and G. panel D shows higher signal with LPS than with C-opsonized virus, but panel G shows that the fluorescence intensities similar.
6. The authors state: "Notably, we identified antibody responses against SARS-CoV-2 as a negative feedback mechanism in limiting complement-induced inflammation via CD32 signalling." Strictly speaking this is not true in that the data presented was a blocking antibody experiment. Can the authors show signaling downstream of IgG binding to CD32 is required for the effect that they are reporting? And, more importantly, how does that signaling block the inflammatory cytokine production that opsonized virus produced in DCs.
7. Can the authors show that levels of C3c and C3d are higher in serum from people with severe COVID-19 than from people with mild disease.

Referee #2:

In this study, Marta Bermejo-Jambrina et al show that complement-opsonized SARS-CoV-2 induced DC maturation, activation of type-I IFN responses, and pro-inflammatory cytokines, via complement receptors CR3/CD11b and CR4/CD11c. In addition, anti-SARS-CoV-2 antibodies abrogated complement-induced DC activation and subsequent type I IFN and cytokine responses via CD32 activation, suggesting a mechanism to restore homeostasis.

The study has several important findings which reveal mechanisms underlying the involvement of complement in immune

responses in COVID-19. However, the quality of the experimental approaches should be increased by including a series of key control conditions in the experiments performed. This would increase the soundness of this paper. The experimental methods should be more precise and detailed.

Major points:

- the experiments of complement-opsonization should be combined with experiments with heat-inactivated serum (HIS), to demonstrate the specificity of the findings. A dose-response curve of serum should be provided. From methods, the concentration of serum (1:10 ratio) used in the assays is not clear.
- In Fig 1A, please specify in the legend that this is a pseudovirus. The experiment of opsonization with SARS-CoV-2 should be shown in the main figures.
- In the experiments with DCs, it would be important to show the conditions of opsonization with HIS, as well as the treatment of DCs with the same percentage of human serum, without the virus.
- Experiments of Fig 1-3 should be performed in the presence of an irrelevant isotype control mAb.
- The experiment with mannan is unclear. Which concentrations of mannan and NHS were used? A dose response curve should be shown. HIS should be included.
- Figure 4: Did the presence of specific anti-SARS-CoV-2 Abs modify complement opsonization? An irrelevant isotype control should be added in the experiment. Since CD16 is also expressed, what is the effect of anti-CD16 in this experiment?
- Figure 6: in this experiment, it would be important to assess plasma levels of complement molecules (C3 at least) in sera from severe and mild COVID patients, as well as the titration of specific anti-SARS-CoV-2 mAbs. The sentence at line 435 "These results suggest that in severe COVID-19 patients, complement is fully activated" is not convincing without a measure of complement molecules. How many sera per condition were tested? Which serum concentration was used?

Referee #3:

General summary and opinion

The manuscript by Bermejo-Jambrine et al. describes a role for complement in the induction of innate and adaptive immunity to SARS-CoV2. The authors show that complement-opsonized SARS-CoV-2 can efficiently interact with dendritic cells via CD11 b/c. Binding induces type I Interferon and pro-inflammatory cytokine responses suggesting a central role for complement in inducing immunity via DCs in the acute phase of SARS-CoV 2 infection. However, the serum from convalescent COVID-19 patients as well as monoclonal antibodies against SARS-CoV-2 attenuate the immune response by complement-opsonized SARS-CoV-2. This effect is mediated by CD32. The results suggest possible therapeutic options.

The pDC are likely essential for a successful immune response to SARS-CoV-2 because of their ability to produce large amounts of type I IFN. SARS-CoV-2 has been demonstrated to infect moDC but the infection has been shown to be abortive. The role and functions of DCs during SARS-CoV-2 infection have not been fully understood. Therefore, it is of relevance to understand in details how SARS-CoV-2 interacts and affects these cells.

The authors have previously published results closely related to the topic of the present manuscript.

The manuscript is interesting and its relevance high enough to justify publishing the data presented here. However, some changes should be addressed to make the message clearer to the reader. In addition, the manuscript contains several inaccuracies, especially in the figures, that should be rectified.

Major points

The authors conclude from their results that the antibody response against SARS-CoV-2 acts as a negative feedback mechanism in limiting complement-induced inflammation via CD32 and that overactivation of the complement system contributes to pathophysiology of severe Covid-19 disease.

However, several previous works have reported that SARS-CoV-2-specific IgG depends on COVID-19 severity and the highest virus-specific IgG antibody titers are observed in severe/critical cases. Aberrant inflammation observed in critical cases is often accompanied by high titers of specific antibodies. So why are specific IgG not able to dampen the immune response by complement-opsonized SARS-CoV-2 in these cases? Role for IgG glycosylation? Timing of the IgG response? The authors should discuss on these aspects and cite relevant references.

In addition, the authors state several times and refer to previous publications showing that DC are not infected by SARS CoV-2.

However they don't show it here for C-opsonized SARS-CoV-2. Uptake of C-opsonized SARS-CoV-2 even without further replication (abortive infection) could also stimulate DC i.e. via intracellular TLRs. The authors should possibly present data showing no uptake of C-opsonized SARS-CoV-2 and discuss this aspect in light of appropriate references.

Minor points

Title: Instead of antibodies, it would be more appropriate to use the term IgG. In fact, the authors show that the effect is mediated by CD32, which is a Fc-receptor for IgG. Please modify accordingly in the title as well as in the main text, Figure title, figure legend etc...

Lines 130,131: please use the term monoclonal antibodies for COVA1-18 und COVA1-27 here, and whenever referring to these mAbs in the main text, Figure title, figure legend etc...

References: van der Donk, et al., Eur J Immunol, 2022. 52(4): p. 646-655 is cited twice (#30 and #33)

Legend to Fig 1: Mention CD11b/c treatment shown in panels A-C.

Fig 1:

1) Fig 1B: binding of SARS-Cov 2 (not opsonized) is probably mediated by heparan sulphates. Why is binding of C-opsonized SARS + anti-CD11b/c even lower than binding of SARS-CoV-2? Please clarify and discuss this finding

2) Is LPS mediated activation of DC also dependent on CD11 b/c or rather TLR4? Please comment and discuss on this aspect and possibly include LPS control +/- CD11b/c for Fig 1 C-E

3) Fig 1F: the mean values of CD80/CD86 expression are almost identical here and do not reflect what shown in fig 1 C, D. Please clarify this point.

Legend to Fig 3: Legend title does not correctly describes what is shown in the figure

Fig 3: The number of dots in figure 3C and 3D is not the same for all conditions

Legend to Fig 4:

1) The condition Ab/C-opsonized SARS-CoV-2 in not mentioned

2) Please mention in the figure legend that sera from mild/moderate cases were used here, as stated in the main text.

Fig 4: IFN-beta is not shown here. Why?

Legend to Fig 5:

1) please change anti-SARS CoV-2 antibodies into anti-SARS CoV2 monoclonal antibodies

2) Line 866: what do you mean with blocks? DC were just exposed to SARS-CoV2. No other conditions have been tested here

3) LPS stimulation is not shown in figure 5 but is mentioned in the figure legend

Fig5: IL12p35 is not shown. Why?

Fig 6B:

1. APOBEC3G, CXCL10, IL-10 and IL12p35 are not shown here. Why?

2. The differences observed between SARS-Cov-2 and C-opsonized SARS-CoV-2 or Ab/C opsonized SARS-CoV-2 is not so clear for IFN-beta. Significance? Please clarify this point.

3. Bar colours are confusing : colour for "sera-opsonized SARS-CoV-2" is the same as for "Unstim." as depicted in the legend but is not in the figure itself. Bar colour for SARS-CoV2 is not consistent in all panels

Discussion

There are two relevant publications closely related to the topic of the manuscript but dealing with different virus systems which should discussed and cited :

Crisci et al J Virol 2016; 29:4939

Tjomsland et al PLoS One 2011; 6:e23542

Point-by-Point response to reviewers' concerns**Referee #1:**

The authors investigated the effect of complement, anti-SARS-CoV-2 antibodies, and cell surface receptors on activation of monocyte-derived DCs by SARS-CoV-2 virus, or by lentiviral vectors pseudotyped with Spike. SARS-CoV-2 association with DCs, and activation of DCs to produce inflammatory cytokines and IL-1, was increased by preincubation of virions with pre-COVID-19 normal human serum (NHS). This activity was attributed to the accumulation of C3c and C3d from the serum on the virus surface. DC activation was blocked by antibodies to the DC cell surface complement receptors CR3 and CR4, and by mannan, an inhibitor of the lectin pathway. The authors then showed that convalescent serum containing anti-SARS-CoV-2 antibodies, as well as anti-Spike monoclonal antibodies, blocked complement-mediated DC activation. Inhibition of DC activation by convalescent sera or by anti-Spike monoclonal antibodies was reversed by anti-FcγRII (CD32) antibody. When serum from people with mild COVID-19 was compared with that from people with severe COVID-19, correlation was shown between disease severity and degree of virion opsonization, indicating that COVID-19 severity correlated with greater complement activation.

Overall, this manuscript describes interesting links between SARS-CoV-2 opsonization by serum complement and innate immune activation of dendritic cells, and correlation between these observations and disease severity. The experiments presented are in themselves reasonably convincing though in most experiments, the authors have derived conclusions based on a single assay. The manuscript would be strengthened if the authors buttressed their findings with some orthogonal experimental approaches and controls. A few suggestions are provided here, along with some minor technical comments and questions:

1. Of the various complement proteins present in sera, why did the authors focus on C3c and C3d for their opsonization experiments. Might other complement components like C3b contribute to opsonization?

We focused on C3c and C3d as these are the final products from C3b activation. As requested, we have now included new data showing that SARS-CoV-2 is also opsonized by C3b and discuss this in the text (Supp. Figure 1B; line 372-375).

2. The authors claim that opsonization and DC activation was due to C3c and C3d present in sera. But sera is a complex mixture of proteins and they only showed that C3c and C3d were associated with the virions. Is there an experiment that can be done to show that C3c and/or C3d are required for opsonization activity and DC activation? For example, does purified C3c and/or C3d opsonize? Alternatively, does depletion of C3c and/or C3d from serum disrupt opsonization?

We thank the reviewer for the suggestion. C3c and C3d derive from C3 activation and therefore cannot be depleted from serum. However, we have now performed opsonization and DC activation with C3-depleted as well as heat inactivated serum. These data show that C3 is required for C3b and C3c opsonization of SARS-CoV-2 as well as activation of dendritic cells. We have included the data and discuss our findings in the text (Figure 1B and Supp. Figure 1B, 2A-C, 4B-F; line 360-361; 372-375; 382-383; 406-408; 534-535).

3. Fig 1A: The authors claim that opsonization requires Spike but they do not have a control to show that opsonization of lentivectors requires Spike. Does complement opsonize lentivectors in the absence of Spike or pseudotyped with other viral glycoproteins? Are the experiments in Fig 1A statistically significant?

As requested, we have performed opsonization experiments with lentivirus particles lacking Spike envelope glycoprotein from SARS-CoV-2. Notably, no opsonization was observed of the lentivirus particles without Spike. We have included these new data in the manuscript (Supp. Figure 1A ; line 361-363; 535-537). We have included statistics for Fig 1A.

4. Fig 1B: Complement seems to increase virus binding to DCs by ~2-fold. This raises a few questions:

- Is binding in the absence of complement due to heparan sulfate?

Indeed, we have previously shown that SARS-CoV-2 binds to DCs via heparan sulfate proteoglycans (Bermejo-Jambrina, Eder et al. 2021). We have clarified this further in the text (line 370-372).

- If so, why do anti-CD11b/c antibodies decrease binding to levels that are lower than seen in the absence of complement?

Opsonized SARS-CoV-2 interacts with CD11b and CD11c but the opsonization might prevent interaction with Heparan Sulfate Proteoglycans and this would result in a lower binding when blocking CD11b and CD11c. We have now discussed this in the text (line 367-372).

- What would be the effect of non-specific antibody or of antibody against another DC cell surface molecule?

- As requested, we have now included an isotype antibody against another DC cell surface molecule and the isotype antibody did not affect the binding of and DC activation by opsonized SARS-CoV-2. We have included the data in the manuscript (Supp. Figure 2D-F and 4B-D; line 384-386, 404-406).

5. Fig 1D and G: There seems to be a discrepancy between panels D and G. panel D shows higher signal with LPS than with C-opsonized virus, but panel G shows that the fluorescence intensities similar.

We apologize for the discrepancy. Fig 1D displays the average of 12 donors and we had selected a donor with high expression of CD86 after opsonized SARS-CoV-2 exposure. We have now selected a more representative donor for panel G-I (Figure 1G-I).

6. The authors state: "Notably, we identified antibody responses against SARS-CoV-2 as a negative feedback mechanism in limiting complement-induced inflammation via CD32 signalling." Strictly speaking this is not true in that the data presented was a blocking antibody experiment. Can the authors show signaling downstream of IgG binding to CD32 is required for the effect that they are reporting? And, more importantly, how does that signaling block the inflammatory cytokine production that opsonized virus produced in DCs.

We thank the reviewer for this question and we have changed the sentence to reflect the role of CD32 binding but not signaling (line 522). We are very interested in the mechanism of how CD32 signaling blocks the inflammatory responses but we feel that identifying the signaling mechanism is beyond the scope of the manuscript.

7. Can the authors show that levels of C3c and C3d are higher in serum from people with severe COVID-19 than from people with mild disease.

We thank the reviewer for the comment. We have now measured C3a levels in serum from mild and severe COVID-19 patients compared to healthy donors. C3a is a cleavage fragment from C3, equivalent to C3b. Moreover, we also measured C5a another potent mediators of inflammation upon complement activation. We have included these data and discuss the findings in the text (Supp. Figure 6B-C; line 338-341; 486-489).

Referee #2:

In this study, Marta Bermejo-Jambrina et al show that complement-opsonized SARS-CoV-2 induced DC maturation, activation of type-I IFN responses, and pro-inflammatory cytokines, via complement receptors CR3/CD11b and CR4/CD11c. In addition, anti-SARS-CoV-2 antibodies abrogated complement-induced DC activation and subsequent type I IFN and cytokine responses via CD32 activation, suggesting a mechanism to restore homeostasis.

The study has several important findings which reveal mechanisms underlying the involvement of complement in immune responses in COVID-19. However, the quality of the experimental approaches should be increased by including a series of key control conditions in the experiments performed. This would increase the soundness of this paper. The experimental methods should be more precise and detailed.

Major points:

- the experiments of complement-opsonization should be combined with experiments with heat-inactivated serum (HIS), to demonstrate the specificity of the findings. A dose-response curve of serum should be provided. From methods, the concentration of serum (1:10 ratio) used in the assays is not clear.

As requested we have now performed experiments with heat-inactivated serum as well as a dose-response curve. We have also performed experiments with C3-depleted serum. Heat inactivation as well as C3 depletion abrogated opsonization of SARS-CoV-2 and SARS-CoV-2 treated with heat inactivated or C3-depleted serum did not induce DC activation nor cytokine induction. We have included the data and discuss our findings in the text. Dose-response curves have been included (Suppl. Figure 3A-F). Moreover we have clarified the use of the concentration of serum in the methods section (Figure 1B and Supp. Figure 1B, 2A-C, 3A-F, 4B-F; line 139-141; 360-361; 372-374; 382-383; 406-408; 535-536).

- In Fig 1A, please specify in the legend that this is a pseudovirus. The experiment of opsonization with SARS-CoV-2 should be shown in the main figures.

As requested, we have specified the use of the pseudovirus in the legend of Fig. 1A. Moreover, the opsonization of SARS-CoV-2 has now been included as a main figure.

- In the experiments with DCs, it would be important to show the conditions of opsonization with HIS, as well as the treatment of DCs with the same percentage of human serum, without the virus.

As requested, we have performed DC stimulations with SARS-CoV-2 treated with C3-depleted as well as heat inactivated serum. These data support our hypothesis that C3-mediated opsonization of SARS-CoV-2 leads to immune activation (Figure 1B and Supp. Figure 1B, 2A-C, 4B-F; 360-361; 372-374; 382-383; 406-408; 535-536).

Moreover, we have included sera alone (Supp. Figure 2G; line 389-390).

- Experiments of Fig 1-3 should be performed in the presence of an irrelevant isotype control mAb.

As requested, we have performed the relevant experiments with an isotype (DC activation and cytokine induction/secretion) (Supp. Figure 2D-F and 4B-D; line 384-386, 404-406).

- The experiment with mannan is unclear. Which concentrations of mannan and NHS were used? A dose response curve should be shown. HIS should be included.

We apologize for the confusion. We have used the concentration of serum (1:10 ratio) as in the rest of the experiments. The mannan concentration used for the experiment was 100µg/mL and we have clarified this in the legends (line 1092). We have now included data with mannan and heat inactivated sera. We have now included data with regard to the heat inactivated sera and have clarified the concentrations used (Supp Figure. 4E-F; line 426-430). We have included a dose response curve of NHS (Supp. Fig. 3A-F; line 389-390).

-Figure 4: Did the presence of specific anti-SARS-CoV-2 Abs modify complement opsonization?

An irrelevant isotype control should be added in the experiment. Since CD16 is also expressed, what is the effect of anti-CD16 in this experiment?

As requested, we have performed a new opsonization ELISA assay including in which we titrated the NHS for concentration-dependent complement opsonization and simultaneously we added a specific anti-SARS-CoV-2 mAb, COVA1-18, at the concentration (0.05ug/ml) used in our assays. Thus, the data showed no change on the complement deposition at different concentration when specific anti-SARS-CoV-2 was present (Supp Figure. 6A; line 484-486).

Moreover, we have also performed cytokine expression in presence of CD16 antibodies. CD16 antibodies did not restore type I IFN expression by DCs treated with Ab/Complement-opsonized SARS-CoV-2. We have included the new data in the manuscript (Supp. Figure. 7B; line 505-506).

- Figure 6: in this experiment, it would be important to assess plasma levels of complement molecules. The sentence at line 435 "These results suggest that in severe COVID-19 patients, complement is fully activated" is not convincing without a measure of complement molecules.

We have now measured C3a levels in serum from mild and severe COVID-19 patients compared to healthy donors. C3a is a cleavage fragment from C3, equivalent to C3b. Moreover, we also measured C5a another potent mediators of inflammation upon complement activation. We have included these data and discuss the findings in the text (Supp. Figure 6B-C; line 338-341; 486-489).

How many sera per condition were tested? Which serum concentration was used?

We apologize for the omission and have now included the numbers (n=5 for mild and n=5 for severe) and concentrations (line 255-258; 474-476; 493-494).

Referee #3:

General summary and opinion

The manuscript by Bermejo-Jambrine et al. describes a role for complement in the induction of innate and adaptive immunity to SARS-CoV2. The authors show that complement-opsonized SARS-CoV-2 can efficiently interact with dendritic cells via CD11 b/c. Binding induces type I Interferon and pro-inflammatory cytokine responses suggesting a central role for complement in inducing immunity via DCs in the acute phase of SARS-CoV 2 infection. However, the serum from convalescent COVID-19 patients as well as monoclonal antibodies against SARS-CoV-2 attenuate the immune response by complement-opsonized SARS-CoV-2. This effect is mediated by CD32. The results suggest possible therapeutic options.

The pDC are likely essential for a successful immune response to SARS-CoV-2 because of their ability to produce large amounts of type I IFN. SARS-CoV-2 has been demonstrated to infect moDC but the infection has been shown to be abortive. The role and functions of DCs during SARS-CoV-2 infection have not been fully understood. Therefore, it is of relevance to understand in details how SARS-CoV-2 interacts and affects these cells.

The authors have previously published results closely related to the topic of the present manuscript.

The manuscript is interesting and its relevance high enough to justify publishing the data presented here. However, some changes should be addressed to make the message clearer to the reader. In addition, the manuscript contains several inaccuracies, especially in the figures, that should be rectified.

Major points

The authors conclude from their results that the antibody response against SARS-CoV-2 acts as a negative feedback mechanism in limiting complement-induced inflammation via CD32 and that overactivation of the complement system contributes to pathophysiology of severe Covid-19 disease.

However, several previous works have reported that SARS-CoV-2-specific IgG depends on COVID-19 severity and the highest virus-specific IgG antibody titers are observed in severe/critical cases. Aberrant inflammation observed in critical cases is often accompanied by high titers of specific antibodies. So why are specific IgG not able to dampen the immune response by complement-opsonized SARS-CoV-2 in these cases? Role for IgG glycosylation? Timing of the IgG response? The authors should discuss on these aspects and cite relevant references.

- We thank the reviewer for the comment and we have now included a discussion about these aspects including references (line 600-614).

In addition, the authors state several times and refer to previous publications showing that DC are not infected by SARS CoV-2. However they don't show it here for C-opsonized SARS-CoV-2. Uptake of C-opsonized SARS-CoV-2 even without further replication (abortive infection) could also stimulate DC i.e. via intracellular TLRs. The authors should possibly present data showing no uptake of C-opsonized SARS-CoV-2 and discuss this aspect in light of appropriate references.

We thank the reviewer for the comment. We have now included data showing that neither SARS-CoV-2 nor complement-opsonized SARS-CoV-2 productively infects DCs. Moreover, we have now included a discussion regarding triggering of intracellular pattern recognition receptors. We have previously shown that DCs neither become infected nor activated by SARS-CoV-2, whereas ectopic expression of ACE-2 leads to infection and activation by cytosolic pattern recognition receptors (van der Donk, Eder et al. 2022). These data strongly suggest that SARS-CoV-2 uptake does not trigger

endosomal TLRs but that replication is required to trigger cytosolic pattern recognition receptors. Indeed, SARS-CoV-2 has been shown to trigger cytosolic Rig-I like receptors (Thorne, Reuschl et al. 2021). Both SARS-CoV-2 and complement-opsonized SARS-CoV-2 are internalized, suggesting that endosomal TLRs are not involved in the observed immune activation by complement-opsonized virus. We have included the data in the manuscript and discuss the findings also in light of other studies (Figure 2A-B line 288-296; 391-393; 560-566).

Minor points

Title: Instead of antibodies, it would be more appropriate to use the term IgG. In fact, the authors show that the effect is mediated by CD32, which is a Fc-receptor for IgG. Please modify accordingly in the title as well as in the main text, Figure title, figure legend etc...

We thank the reviewer for the comment. We agree that CD32 is a Fc-receptor for IgG, but since we use also patient sera, which contains several kind of antibodies, we prefer to keep the term antibodies instead of specify IgG.

Lines 130,131: please use the term monoclonal antibodies for COVA1-18 und COVA1-27 here, and whenever referring to these mAbs in the main text, Figure title, figure legend etc... References: van der Donk, et al., Eur J Immunol, 2022. 52(4): p. 646-655 is cited twice (#30 and #33). Legend to Fig 1: Mention CD11b/c treatment shown in panels A-C.

We thank the reviewer for noticing the mistakes and omissions. We have corrected these in the manuscript.

Fig 1:

1) Fig 1B: binding of SARS-CoV 2 (not opsonized) is probably mediated by heparan sulphates. Why is binding of C-opsonized SARS + anti-CD11b/c even lower than binding of SARS-CoV-2? Please clarify and discuss this finding.

Indeed, SARS-CoV-2 interacts with heparan sulfate proteoglycans whereas our data show that opsonized SARS-CoV-2 interacts with CD11b and CD11c. The opsonization might prevent interaction with heparan sulfated proteoglycans and this would result in a lower binding when blocking CD11b and CD11c. We have now discussed this in the text (line 367-372).

2) Is LPS mediated activation of DC also dependent on CD11 b/c or rather TLR4? Please comment and discuss on this aspect and possibly include LPS control +/- CD11b/c for Fig 1 C-E.

We apologize for the confusion. LPS is included as a control for DC activation and cytokine responses. LPS triggers TLR4 but does not interact with CD11b and CD11c. We therefore did not include data with the LPS control and CD11b/c antibodies.

3) Fig 1F: the mean values of CD80/CD86 expression are almost identical here and do not reflect what shown in fig 1 C, D. Please clarify this point.

We apologize for the discrepancy. Fig 1D displays the average of 12 donors and we had selected a donor with high expression of CD86 after opsonized SARS-CoV-2 exposure. We have now selected a more representative donor for panel G (Figure 1G-I).

Legend to Fig 3: Legend title does not correctly describes what is shown in the figure

We have now corrected the legend title.

Fig 3: The number of dots in figure 3C and 3D is not the same for all conditions.

We apologize for the mistake. We have now corrected in the figure and in the figure legend accordingly.

Legend to Fig 4:

1) The condition Ab/C-opsonized SARS-CoV-2 is not mentioned

2) Please mention in the figure legend that sera from mild/moderate cases were used here, as stated in the main text.

We have corrected the mistakes and omissions.

Fig 4: IFN-beta is not shown here. Why?

We apologize for the omission. We have now included the IFN-beta data (Fig. 4A; line 442-443, 445-447).

Legend to Fig 5:

1) please change anti-SARS CoV-2 antibodies into anti-SARS CoV2 monoclonal antibodies

2) Line 866: what do you mean with blocks? DC were just exposed to SARS-CoV2. No other conditions have been tested here

3) LPS stimulation is not shown in figure 5 but is mentioned in the figure legend

We apologize for the mistake. We have corrected now the figure legend accordingly.

Fig5: IL12p35 is not shown. Why?

We have now included the IL-12p35 data in the figure and discussed in the text (Fig. 5G; lines 461-463, 463-464).

Fig 6B:

1. APOBEC3G, CXCL10, IL-10 and IL12p35 are not shown here. Why?

We apologize for the confusion. We have now included the data (Supp. Fig. 6D-G) and discussed the findings (Line 500-506) and also investigated CD16 involvement (Supp. Figure 7A-B).

2. The differences observed between SARS-Cov-2 and C-opsonized SARS-CoV-2 or Ab/C opsonized SARS-CoV-2 is not so clear for IFN-beta. Significance? Please clarify this point.

We have now clarified the findings in more detail regarding IFN-beta (line 494-496).

3. Bar colours are confusing : colour for "sera-opsonized SARS-CoV-2" is the same as for "Unstim." as depicted in the legend but is not in the figure itself. Bar colour for SARS-CoV2 is not consistent in all panels

We apologize for the colors, we have now corrected in the figures, SARS-CoV-2 bars and sera-opsonized SARS-CoV-2 (light grey).

Discussion

There are two relevant publications closely related to the topic of the manuscript but dealing with different virus systems which should be discussed and cited :

Crisci et al J Virol 2016; 29:4939

Tjomsland et al PLoS One 2011; 6:e23542

We thank the reviewer for the suggestion. We have now cited and discussed those studies (line 549-552).

Dear Prof. Geijtenbeek,

Thank you for the submission of your revised manuscript to The EMBO Journal. We have now received the comments of the three referees that were asked to re-evaluate your study (included below). As you will see, the referees are satisfied with the revision, acknowledge that the previous concerns have been largely addressed and that thorough responses have been provided to their questions, and they now support publication once a few minor remaining issues are addressed:

- clearer interpretation/discussion of the role and mechanism of action of CD32 in the processes presented in this study is needed (ref. #2)
- all language errors in the new parts added during revision should be corrected (ref. #2)
- in the title of the legend of Fig. 4 "monoclonal" should be deleted (ref. #3).

Please address these minor issues in a revised version of your manuscript and describe all changes in a point-by-point response.

From the editorial side, there are also a few things that we need from you before we can proceed with acceptance of the manuscript:

- Please enter all relevant funding information in our online manuscript handling system. It should match exactly the information provided in the Acknowledgements section of your manuscript. We noticed that the "European Research Council (Advanced grant 670424)" is currently missing from our online system.
- Please provide a list of up to 5 keywords after the abstract of your revised manuscript.
- Please note our reference format: <https://www.embopress.org/page/journal/14602075/authorguide#referencesformat>. It is alphabetical (not numbered), and the names of the first 10 authors of each publication (followed by "et al." if there are more than 10 co-authors) should be provided.
- Please note that a data availability statement is mandatory. If your study does not include any datasets requiring deposition in a public database, please add the statement: "This study includes no data deposited in external repositories." under the heading "Data availability" at the end of Materials and Methods.
- Please change the heading of your conflict-of-interest statement to: "Disclosure and competing interests statement".
- The author contributions statement should be removed from the manuscript file. Instead, we now use the CRediT system to specify the contributions of each author in the journal submission system. Please use the free text box to provide more detailed descriptions. See also our guide to authors: <https://www.embopress.org/page/journal/14602075/authorguide#authorshipguidelines>.
- According to our journal's policy, "data not shown/published" (stated twice on page 11 of your manuscript) is not permitted. All data referred to in the paper should be displayed in the main or Expanded View figures, or in the Appendix. Please add these data or change the text accordingly if these data are not central to the study and its conclusions.
- The callouts for Expanded View (EV) Figures should be Figure EV# instead of Supplementary Figure #. Please update both their legends and their callouts throughout the manuscript accordingly.
- In your Author Checklist you have indicated that you have used a select agent in the "Dual Use Research of Concern (DURC)" section. If the study is subject to dual use research of concern regulations, this should be indicated in this section, and the name of the authority granting approval, as well as the reference number for the regulatory approval should be provided in the manuscript. The section in your Checklist should also be completely filled out. Please see the Checklist for more information and the link to biosecurity documents and list of select agents and toxins.
- Please re-organize your Source Data to one zipped file/folder per figure. For example, all Source Data files for Figure 1 need to be saved in a single folder and this needs to be zipped and then uploaded as "SD Figure 1.zip". For EV and/or Appendix Figures, please zip all their source data together in a single folder.
- The manuscript sections are in the wrong order. Please follow the order: Title page, Abstract, Keywords, Introduction, Results, Discussion, Materials and Methods, Data availability, Acknowledgements, Disclosure and competing interests statement, References, Figure legends, Expanded View Figure legends.
- Please note that The EMBO Journal papers are accompanied online by:
A) a short (1-2 sentences) summary of the findings and their significance,
B) 2-5 short bullet points highlighting the key results, and
C) a synopsis image that is exactly 550 pixels wide and 300-600 pixels high (the height is variable). You can either show a model

or key data in the synopsis image. Please note that the text needs to be readable at the final size. Please upload this information along with your revised manuscript (the text for A and B should be provided in a separate Word file).

- The legend for figure 4g is incorrectly labelled as 4f in the legend. This needs to be rectified.
- The legend for figure 5f is incorrectly labelled as 5g in the legend. This needs to be rectified.
- In the legend of supplementary figure 3, the statistical information for figure panels 3a, c, e is incorrectly labelled as 3a-c. This needs to be rectified.
- In figures 1c; 3c-d; 4a-g; 6c-d; supplementary figures 1c-d; there is a mismatch between the annotated p values in the figure legend and the annotated p values in the figure file that should be corrected.
- For the supplementary figures 2b, d; 4b; 6f; p-values and statistical tests are indicated in the legends. However, comparison for the same, ""**/*"" has not been represented in the figures. Please rectify this in the figures or legends as applicable.
- Please note that information related to n is missing in the legend of figure 6a.

When these issues are resolved, I will contact you again to discuss with you a few suggestions for minor textual improvements in the title, abstract and synopsis text.

Please also note that as part of the EMBO publications' Transparent Editorial Process, The EMBO Journal publishes online a Peer Review File along with each accepted manuscript. This File will be published in conjunction with your paper and will include the referee reports, your point-by-point response and all pertinent correspondence relating to the manuscript. You can opt out of this by letting the editorial office know (contact@embojournal.org). If you do opt out, the Peer Review File link will point to the following statement: "No Peer Review File is available with this article, as the authors have chosen not to make the review process public in this case."

We look forward to seeing a final version of your manuscript as soon as possible. Please use this link to submit your revision: <https://emboj.msubmit.net/cgi-bin/main.plex>

Yours sincerely,

Referee #1:

The authors have provided thoughtful and thorough responses to all of the comments in my initial review.

Referee #2:

The authors addressed most of my questions. However, there are still some points unclear to me, that could be interpreted in a more critical manner.

In particular, the role and mechanism of action of CD32 in the processes presented in this study are not clearly dissected or interpreted.

There are several language errors in the new parts added in the revised manuscript.

Referee #3:

I appreciate the effort of the authors and their willingness to perform additional analyses and to provide appropriate edits and explanations. With the additional data and editing work, I believe that my prior concerns and questions have been mostly addressed.

One minor point: in the title of legend to Fig 4 "monoclonal" should be deleted.

Point-by-Point response to reviewers' concerns**Referee #2:**

The authors addressed most of my questions. However, there are still some points unclear to me, that could be interpreted in a more critical manner.

In particular, the role and mechanisms of action of CD32 in the processes presented in the study are not clearly dissected or interpreted.

-We thank the reviewer for the comment and we have now included a discussion about these aspects including references (line 355-378).

There are several language errors in the new parts added in the revised manuscript.

We thank the reviewer for pointing out languages mistakes in the manuscript. We have revised the manuscript and correct them.

Referee #3:

I appreciate the effort of the authors and their willingness to perform additional analyses and to provide appropriate edits and explanations. With the additional data and edition work, I believe that my prior concerns and questions have been mostly addressed.

One minor point: in the title of legend of Figure 4 "monoclonal" should be deleted.

-We thank the reviewer for the kind comment. We apologize for the mistake and we have now removed it from the figure 4 title legends.

Editorial comments:

Please enter all relevant funding information in our online manuscript handling system. It should match exactly the information provided in the Acknowledgements section of your manuscript. We noticed that the „European Research Council (Advanced grant 670424)“ is currently missing from our online system.

- As requested, we have now matched the information in the manuscript and online system.

Please provide a list of up to 5 keywords after the abstract of your revised manuscript.

- We have now included 5 keywords.

Please note our reference format.

- We have changed the format of the references accordingly.

Please note that a data availability statement is mandatory.

-We have included the statement in the text.

Please change the heading of your conflict-of-interest statement to: "Disclosure and competing interests statement".

- We have changed the statement accordingly.

The author contributions statement should be removed from the manuscript file. Instead, we now use the CRediT system to specify the contributions of each author in the journal submission system. Please use the free text box to provide more detailed descriptions.

- We have removed author contributions statement and specified the contributions in the submission system.

According to our journal's policy, "data not shown/published" (stated twice on page 11 of your manuscript) is not permitted. All data referred to in the paper should be displayed in the main or Expanded View figures, or in the Appendix. Please add these data or change the text accordingly if these data are not central to the study and its conclusions.

- We have changed the text accordingly and removed the data not shown statements as the

data are not central to the study and its conclusions. The callouts for Expanded View (EV) Figures should be Figure EV# instead of Supplementary Figure #. Please update both their legends and their callouts throughout the manuscript accordingly.

- We have updated the legends and their callouts.

In your Author Checklist you have indicated that you have used a select agent in the "Dual Use Research of Concern (DURC)" section. If the study is subject to dual use research of concern regulations, this should be indicated in this section, and the name of the authority granting approval, as well as the reference number for the regulatory approval should be provided in the manuscript. The section in your Checklist should also be completely filled out. Please see the Checklist for more information and the link to biosecurity documents and list of select agents and toxins.

- We apologise for the mistake but the study is not subject to dual use research of concern as we did not modify SARS-CoV-2 or otherwise modify its infectivity.

Please re-organize your Source Data to one zipped file/folder per figure. For example, all Source Data files for Figure 1 need to be saved in a single folder and this needs to be zipped and then uploaded as "SD Figure 1.zip". For EV and/or Appendix Figures, please zip all their source data together in a single folder.

- We have re-organized our Source Data.

The manuscript sections are in the wrong order. Please follow the order: Title page, Abstract, Keywords, Introduction, Results, Discussion, Materials and Methods, Data availability, Acknowledgements, Disclosure and competing interests statement, References, Figure legends, Expanded View Figure legends.

- We have corrected the order of the sections.

Please note that The EMBO Journal papers are accompanied online by:

- A) a short (1-2 sentences) summary of the findings and their significance,
- B) 2-5 short bullet points highlighting the key results, and
- C) a synopsis image that is exactly 550 pixels wide and 300-600 pixels high (the height is variable). You can either show a model or key data in the synopsis image. Please note that the text needs to be readable at the final size.

Please upload this information along with your revised manuscript (the text for A and B should be provided in a separate Word file).

- We have now provided the necessary information and synopsis figure (Synopsis file).

The legend for figure 4g is incorrectly labelled as 4f in the legend. This needs to be rectified. The legend for figure 5f is incorrectly labelled as 5g in the legend. This needs to be rectified. In the legend of supplementary figure 3, the statistical information for figure panels 3a, c, e is incorrectly labelled as 3a-c. This needs to be rectified. In figures 1c; 3c-d; 4a-g; 6c-d; supplementary figures 1c-d; there is a mismatch between the annotated p values in the figure legend and the annotated p values in the figure file that should be corrected.

For the supplementary figures 2b, d; 4b; 6f; p-values and statistical tests are indicated in the legends. However, comparison for the same, ""**/*"" has not been represented in the figures. Please rectify this in the figures or legends as applicable. Please note that information related to n is missing in the legend of figure 6a.

- We have corrected the text in the figure legends.

Dear Prof. Geijtenbeek,

I am pleased to inform you that your manuscript has been accepted for publication in The EMBO Journal.

Yours sincerely,
